# Multimodal Large Language Models Make Text-to-Image Generative Models Align Better

**Xun Wu**[1], **Shaohan Huang**[1][✉], **Guolong Wang**[2], **Jing Xiong**[3], **Furu Wei**[1]

[1] Microsoft Research Asia,  [2] University of International Business and Economics
[3] The University of Hong Kong

xunwu@microsoft.com, shaohanh@microsoft.com, fuwei@microsoft.com

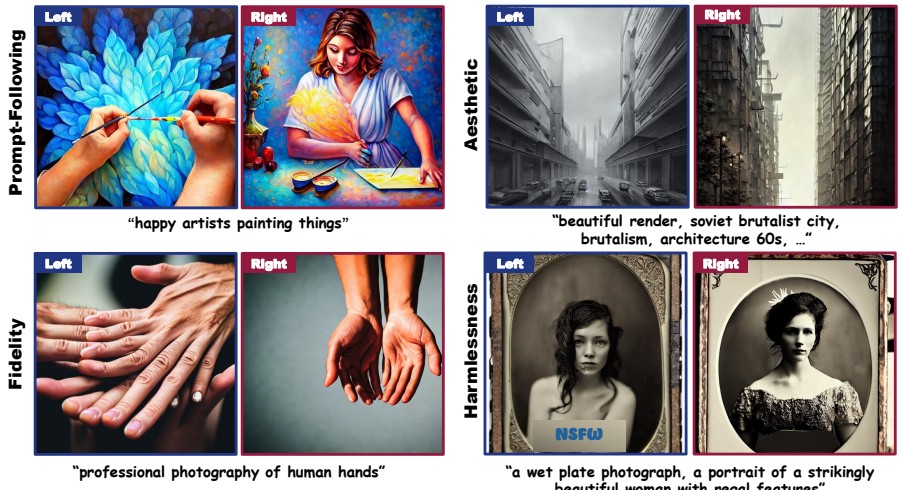

Figure 1: **Fine-grained feedback from multimodal large language model help to yield more human-preferred images**. **Left**: Output generated by the baseline text-to-image generative model. **Right**: Output generated by the baseline model optimized with fine-grained feedback from multimodal large language model. We illustrate improvements in generation quality across four aspects: *Prompt-Following*, *Aesthetic*, *Fidelity* and *Harmlessness*. See in Appendix for more visualization examples.

## Abstract

Recent studies have demonstrated the exceptional potentials of leveraging human preference datasets to refine text-to-image generative models, making it to generate more human-preferred images. Despite these advances, current human preference datasets are either prohibitively expensive to construct or suffer from a lack of diversity in preference dimensions, resulting in limited applicability for instruction tuning in open-source text-to-image generative models and hinder further exploration. To address these challenges, we first leverage multimodal large language models to create **VisionPrefer**, a fine-grained preference dataset that captures multiple preference aspects (prompt-following, aesthetic, fidelity, and harmlessness). Then we train a corresponding reward model, **VP-Score**, over VisionPrefer to guide the tuning of text-to-image generative models. The preference prediction accuracy of VP-Score is validated to be comparable to that of human annotators. To validate the effectiveness of VisionPrefer and VP-Score, we adopt two reinforcement learning methods, Proximal Policy Optimization (PPO) and Direct Policy Optimization (DPO), to supervised fine-tune generative models, and extensive experimental

results demonstrate that VisionPrefer significantly improves text-image alignment in compositional image generation across diverse aspects, e.g., aesthetic, and generalizes better than previous human-preference metrics across various image distributions. Our findings indicates that the integration of AI-generated synthetic data as a supervisory signal is a promising avenue for achieving improved alignment with human preferences in text-to-image generative models. VisionPrefer and VP-Score are available at https://github.com/yushuiwx/VisionPrefer.git .

# 1   Introduction

Text-to-image generative models [23, 21, 19, 26] have experienced rapid advancements in recent years. For example, large-scale text-to-image diffusion models, exemplified by Imagen [24] and DALL·E 2 [19], have demonstrated the capability to generate high-quality and creative images when provided with textual prompts. However, despite recent progress, current generative models still exhibit misalignment with human preferences, such as conflicts with text prompts or incorrect content [17]. A pivotal approach to addressing this issue is utilizing Reinforcement Learning from Human Feedback (RLHF) [16, 27, 1] to supervised fine-tune text-to-image generative models with preference data [13, 3, 5, 17].

Preference data is crucial for aligning generative models with text prompts. However, existing human-crafted preference datasets, such as HPD v2 [33, 32] and Pick-a-Pic [10], either provide only broad, general preference comparisons without fine-grained, accurate preference evaluations, or they are limited in size. Additionally, using humans for preference annotation is expensive and time-consuming, restricting progress in this research area.

Drawing inspiration from recent research utilizing AI-generated preference data as training supervise signal for Large Language Models (LLMs) alignment on Natural Language Processing domain [12, 2], we pose the following question:

*Can Multimodal Large Language Models act as a Human-Aligned Preference Annotator*
*for Text-to-Image Generation?*

These multimodal large language models (MLLMs), trained on web-scale text and text-image pairs, have already demonstrated formidable capabilities in image understanding. To this end, we first introduce VisionPrefer, a publicly available AI-generated dataset that features millions of finely-grained preferences concerning model-generated images. Compared with existing human preference datasets, VisionPrefer offers the following benefits:

- **Scalability & Low Cost**: As shown in Table 1, VisionPrefer encompasses 1.2 M preference choices across 179 K pairs of images, establishing it as the largest text-to-image generation preference dataset to date. Additionally, because VisionPrefer is annotated by MLLMs, it can be easily expanded further and the construction cost is significantly lower than human annotation.
- **Fine-grained preference**: To more accurately and diversely evaluate the preference scores of generated images, we carefully develop a detailed preference annotation guideline for MLLMs, which covers four distinct aspects: *Prompt-Following*, *Fidelity*, *Aesthetic*, and *Harmlessness*. The detail requirement for each aspect is presented at Table 3.
- **Comprehensive feedback formats**: Unlike existing benchmarks that provide only rankings or preference indices, our VisionPrefer not only provides preference rankings but also includes preference scores and textual explanations for the preference annotations from each aspect, which makes VisionPrefer more versatile, e.g., allowing it to serve as a textual guiding resource for image re-editing and refinement.

Building on the VisionPrefer, we conducted an extensive investigation into its most effective utilization. First, we developed a preference reward model named VP-Score optimized on VisionPrefer, trained to evaluate generated images based on their likelihood of being preferred by humans. Experimental results demonstrate that VP-Score exhibits a competitive correlation with human preferences compared to existing human preference reward models. Moreover, we employ two reinforcement learning methods, Proximal Policy Optimization (PPO) and Direct Policy Optimization (DPO), to enhance generative models to better align with human preferences. As illustrated in Figure 1, VisionPrefer markedly enhances text-image alignment in compositional image generation across diverse aspects, such as aesthetics. In summary, our contributions are as follows:

Table 1: Statistics of existing preference datasets for text-to-image generative models. "Fine-grained" denote containing preference regarding multiple aspects or not.

| Dataset | Corresponding Reward Model | Annotator | Prompts | Preference Choices | Open Source? | Fine Grained? | Feedback Format | | |
|---|---|---|---|---|---|---|---|---|---|
| | | | | | | | Ranking | Text | Scalar |
| RichHF-18K [15] | – | Human | 18K | 18K | ✗ | ✓ | ✓ | ✗ | ✗ |
| HPD v1 [33] | HPS v1 | Discord users | 25K | 25K | ✓ | ✗ | ✓ | ✗ | ✗ |
| HPD v2 [32] | HPS v2 | Human Expert | 108K | 798K | ✗ | ✗ | ✓ | ✗ | ✗ |
| ImageRewardDB [34] | ImageReward | Human Expert | 9K | 137K | ✓ | ✓ | ✓ | ✗ | ✗ |
| Pick-a-Pic (v2) [10] | PickScore | Web users | 59K | 851K | ✓ | ✗ | ✓ | ✗ | ✗ |
| VisionPrefer (ours) | VP-Score | GPT-4 V(ision) | **179K** | **1.2M** | – | ✓ | ✓ | ✓ | ✓ |

- We construct VisionPrefer, a large-scale, high-quality, and fine-grained preference dataset for text-to-image generative alignment. Compared with existing preference datasets, VisionPrefer has the advantages of scalability, fine-grained annotations, and comprehensive feedback format.
- Based on VisionPrefer, we propose a reward model, VP-Score, which achieves a competitive correlation with human preferences with other automated human preference metrics.
- Experimental results demonstrate the effectiveness of both VisionPrefer and VP-Score. Additionally, we provide a comprehensive analysis of them to gain a deeper understanding of how AI-generated synthetic data and models trained on such data impact future research in this domain.

## 2  Related Work

**Text-to-Image Generative Models Alignment.** While existing text-to-image generative models often generate images that do not closely match human preferences, thus alignment in the context of diffusion has garnered increasing attention [28, 10, 5, 17, 28]. There are two main types of text-to-image generative models alignment algorithms: (i) *Proximal Policy Optimization (PPO)*. For example, reward weighted method [13] first explores using human feedback to align text-to-image models with human preference. ReFL [34] trains a reward model, ImageReward, using human preferences and subsequently utilizes it for fine-tuning. (i) *Direct Policy Optimization (DPO)*. DPOK [7] fine-tunes text-to-image diffusion models by using policy gradient to maximize the feedback-trained reward. ZO-RankSGD [28] optimizes diffusion in an online fashion with human ranking feedback. RAFT [5] and AlignProp [17] tune the generative model to directly increase the reward of generated images. Several manually annotated preference datasets are proposed to support above algorithms [32, 34, 10]. Their overall statistics are shown in Table 1. These manually annotated data have two drawbacks. First, manual annotation need heavy cost, leading to a small-size set. Second, manual annotations are prone to specific biases [32].

**Reinforcement Learning from AI Feedback.** [2] introduced the idea of Reinforcement Learning from AI Feedback (RLAIF), which used LLM-labeled preferences in conjunction with human-labeled preferences to jointly optimize for the two objectives of helpfulness and harmlessness. Recent works have also explored related techniques for generating rewards from LLMs [20, 11, 35]. These works demonstrate that LLMs can generate useful signals for reinforcement learning fine-tuning. However, RLAIF for text-to-image generative model alignment is less explored. [30] leveraged MLLMs to assess the alignment between generated images and input texts, focusing on aspects like object number and spatial relationship. T2I-CompBench [9] utilized MLLMs like BLIP-VQA to evaluate the text-to-Image generative models. Our work diverges from previous works in two principal ways. (1) Previous approaches provided limited data, sometimes even less than what's annotated manually (e.g., 6K in [9]). (2) Prior methods had limited aspects in alignment, lacking consideration for aspects such as fidelity [30].

## 3  VisionPrefer

We introduce VisionPrefer, a fine-grained preference dataset constructed by collecting feedback from MLLMs annotators. The collection pipeline of VisionPrefer is shown in Figure 2, which mainly consists of three steps: prompt generation, image generation and preference generation.

**Step-1: Prompt Generation.** We generate prompts based on DiffusionDB [29], a large-scale text-to-image prompt benchmark containing 1.5M user-written prompts following two steps: (1) Polish. As discussed in [32], a significant portion of the prompts in the DiffusionDB is biased towards certain

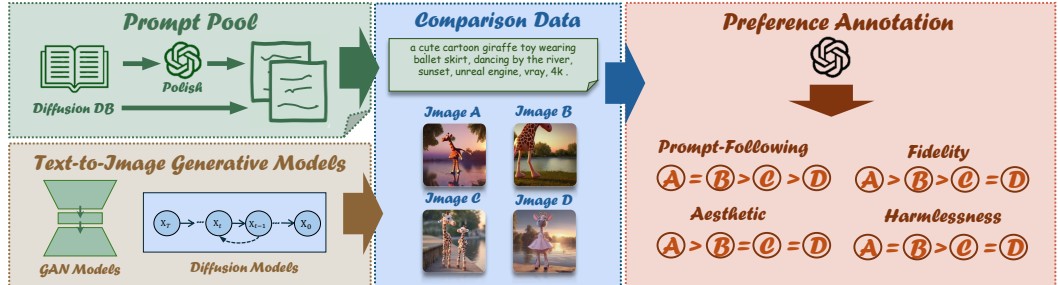

Figure 2: VisionPrefer construction pipeline. We sample textual prompts and text-to-image generative models from pools to generate diverse comparison data, then query GPT-4 V with detailed illustrations for fine-grained and high-quality annotations in both textual and numerical formats.

styles (e.g., "Greg Rutkowski" appears in around 15% of prompts). We utilize GPT-4 to polish prompts in DiffusionDB to obtain unbiased prompts following instructions shown in Appendix F.1. (2) NSFW Filting. We employ state-of-the-art NSFW detectors [8] to compute an NSFW score for each prompt and filter out prompts that exceed a certain threshold, following [29]. After these two steps, we combine both polished prompts and the original prompts in DiffusionDB as our final prompt benchmark, which contains 179K prompts.

**Step-2: Image Generation.** We generate images using different text-to-image generative models (see details in Appendix C.2) by sampling textual prompts constructed in Step-1 as input. For each prompt, we generate four images by randomly selecting different generative models from the model pools with different classifier-free guidance scale values, to ensure high diversity. This diversity allows for a comprehensive evaluation of a preference prediction model's generalization capability and facilitates the training of a more generalizable model. Finally, we obtain 0.716M images.

**Step-3: Preference Generation.** We employ state-of-the-art multimodal large language model, GPT-4 V, to provide three types of feedback: (1) *Scalar scores* that indicate the fine-grained quality regarding multiple aspects, (2) *Preference ranking* according to the scalar scores, and (3) *Textual explanations* that give detailed guidance on how to improve the completion, encompassing four distinct fine-grained aspects namely: *Prompt-Following*, *Aesthetic*, *Fidelity*, and *Harmlessness* for each generated image (See the example in Table 12). Detailed input instructions for GPT-4 V to annotate preference labels are in Appendix F.2. Finally, we obtain 1.2M preference choices.

## 4 Experiments

In this section, we first train a corresponding reward model named VP-Score and evaluate it on existing human-preference datasets (§ 4.1). Next, we enhance existing text-to-image generative models by adopting two reinforcement learning algorithms (§ 4.2) to validate the efficacy of VisionPrefer and VP-Score. After that, we design a simple pipeline to edit generated images with the textual explanations in VisionPrefer(§ 4.3).

### 4.1 Reward Modeling

**Training Setting.** We train the VP-Score over VisionPrefer. VP-Score adopts the same model structure as ImageReward [34], which is a open-source human-preference reward model and utilizes BLIP [14] as the backbone. Similarly to training the reward model for the language model [27, 16], we formulate the preference annotations in VisionPrefer as rankings. Specifically, we employ the average scores of each sample in VisionPrefer across four aspects as the final preference score, and then we have $k$ images ranked generated by the same prompt $\mathbf{T}$ according to final preference score (the best to the worst are denoted as $\mathbf{x}_1 \succ \mathbf{x}_2 \succ ... \succ \mathbf{x}_k$). For each comparison, if $\mathbf{x}_i$ is better and $\mathbf{x}_j$ is worse, the loss function can be formulated as:

$$\text{loss}(\theta) = -\mathbb{E}_{(\mathbf{T},\mathbf{x}_i,\mathbf{x}_j)\sim\mathcal{D}} \left[\log\left(\sigma\left(f_\theta\left(\mathbf{T},\mathbf{x}_i\right) - f_\theta\left(\mathbf{T},\mathbf{x}_j\right)\right)\right)\right] \tag{1}$$

where $f_\theta(\mathbf{T}, \mathbf{x})$ is a scalar value of reward model for prompt $\mathbf{T}$ and image $\mathbf{x}$.

**Evaluation Results.** We evaluate the preference prediction accuracy on the test sets among three human preference datasets: ImageRewardDB [34], HPD v2 [32] and Pick-a-Pic [10]. Furthermore,

Table 2: Preference prediction accuracy across the test sets of ImageRewardDB, HPD v2 and Pick-a-Pic. The Aesthetic Classifier makes prediction without seeing the text prompt. The best performance is in bold, and the second-best performance is underlined.

| Model | ImageRewardDB | HPD v2 | Pick-a-Pic | Average |
|---|---|---|---|---|
| CLIP ViT-H/14 [18] | 57.1 | 65.1 | 60.8 | 60.82 |
| Aesthetic [25] | 57.4 | 76.8 | 56.8 | 62.44 |
| ImageReward [34] | 65.1 | 74.0 | 61.1 | 66.31 |
| HPS [33] | 61.2 | 77.6 | 66.7 | 67.84 |
| PickScore [10] | 62.9 | 79.8 | **70.5** | 70.40 |
| HPS v2 [32] | 65.7 | **83.3** | 67.4 | **71.32** |
| VP-Score | **66.3** | 79.4 | 67.1 | 70.46 |

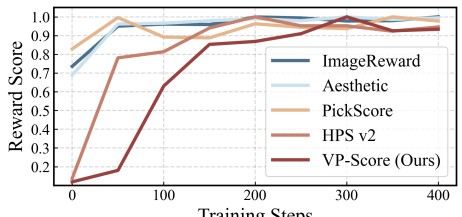

Figure 3: Evaluation results of the text-to-image model's generation quality across multiple reward models when maximizing scores from VP-Score during the PPO training process. All scores are normalized for a better visualization.

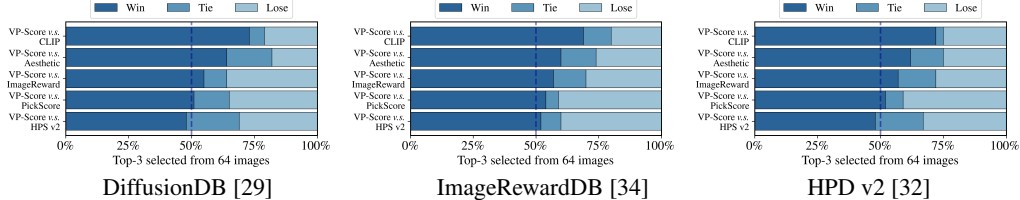

|  DiffusionDB [29] | ImageRewardDB [34] | HPD v2 [32] |
|---|---|---|

Figure 4: Win rates of generative models optimized with VP-Score compared to other reward models on three test benchmarks for PPO experiments. 'Tie' indicates instances where annotators think two images are of comparable quality.

to better demonstrate the model's generalization performance, we computed the harmonic mean of accuracy across three sets for each model as an overall indicator of model performance. We use the CLIP score [18], BLIP score [14], Aesthetic score [25], ImageReward [34], HPS [33], HPS v2 [32] and PickScore [10] as baselines to compare with the VP-Score.

The results are presented at Table 2. Our VP-Score demonstrates strong competitiveness compared to the current state-of-the-art reward models trained on human preference data. It achieves the second-best average performance among all preference reward models, following only HPS v2. Moreover, our model achieves optimal performance on the ImageRewardDB dataset, achieving a 0.6 performance gain compared to HPS v2. These results validate that leveraging fine-grained feedback provided by MLLMs enables learning a proficient human preference reward model.

## 4.2 Fine-tuning Text-to-Image Generative Models

We aim to leverage the constructed preference dataset to align the performance of generative models more closely with human preferences. We utilize two popular reinforcement learning methods for fine-tuning: (1) Proximal Policy Optimization (PPO), where select ReFL [34] as our PPO implementation to adjust the generative model. (2) Direct Preference Optimization (DPO), which allows for direct model fine-tuning using preference data without a reward model, employing D3PO [36] as our DPO implementation. We use Stable Diffusion v1.5 [22] as the text-to-image generative model.

**Training Setting.** For PPO experiments, we randomly sample 20,000 real user prompts from DiffusionDB [29] and 10,000 prompts in ImageRewardDB [34] as the training dataset. We compare VP-Score against five open-source reward models, including ImageReward [34], PickScore [10], and HPS v2 [32], all trained on large-scale preference datasets (see Table 1). All models are fine-tuned with identical data and settings for consistency. For DPO experiments, we compare our VisionPrefer along with three open-source large-scale human-annotated preference datasets, ImageRewardDB [34], HPD [33] and Pick-a-Pic [10] (see in Table 1). Notably, VisionPrefer scores are averaged across four aspects for fine-tuning, with an analysis on individual aspect scores presented in § 5. Both PPO and DPO use the same test benchmarks: 400 real user prompts from DiffusionDB [29], 200 prompts from ImageRewardDB [34] and 400 prompts from HPD v2 [33]. Further details are in Appendix D.

**PPO Results.** First, we visualize the evolution of various metrics as the model training steps increase when using our VP-Score as the reward function at Figure 3. As training progresses, all metrics, including human preference metrics like HPS v2, show an increasing trend. This indicates consistency

**beautiful dark landscape, person wearing a virtual reality headset, intricate, epic lighting, cinematic composition, hyper realistic, 8k resolution, unreal engine 5**

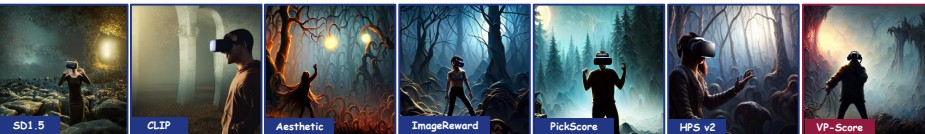

**Profile portrait in Angolan realist style with ultramarine blue, venetian red, and titanium white colors**

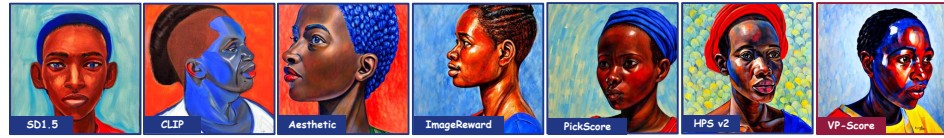

Figure 5: Qualitative results for PPO experiments. SD 1.5 denotes the Stable Diffusion v1.5 model without any fine-tune. See Appendix for more samples.

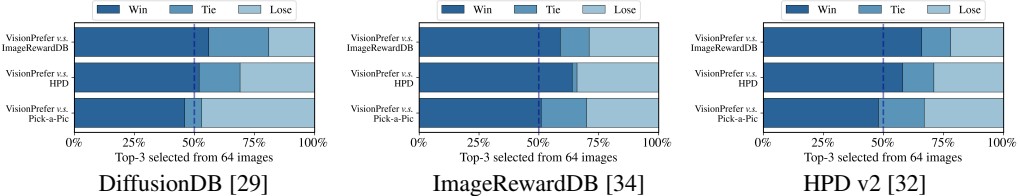

Figure 6: Win rates of generative models optimized with VP-Score compared to other reward models on three test benchmarks for DPO experiments. 'Tie' indicates instances where annotators think two images are of comparable quality.

between our VP-Score and other human preference metrics, demonstrating that VP-Score reliably aligns generative model outputs with human preferences.

Then, we conduct a human preference study. Specifically, we use these fine-tuned generative models to generate 64 images for each prompt in evaluation dataset, followed by a top-3 selection by the corresponding reward models. Finally, ten human annotators rank these selected images. The results are presented at Figure 4, detailed win count and win rates can be found in Table 4. We observed that VP-Score fine-tuned generative model's Win+Tie ratio exceeds 50% when compared to all other models across all three test benchmarks, including some trained on large-scale human preference datasets like HPS v2. This suggests that, compared to other human preference reward model, VP-Score can serve as a reliable and competitive reward model for fine-tuning generative models to produce outputs closer to human preferences. This further underscores the effectiveness and competitiveness of VisionPrefer.

The corresponding qualitative results shown at Figure 5 demonstrate that VisionPrefer fine-tuned generative model can generate images that are more aligned to text and with higher fidelity and avoid toxic contents. More qualitative results can be found in Figure 13.

**DPO Results.** We conduct a human preference study using the same procedure as PPO experiments at Figure 6, detailed win count and win rates can be found in Table 5. We found that the Win+Tie ratio of the generative model optimized on our VisionPrefer, when compared to the other three large-scale human datasets, exceeds 50%, substantiating the competitiveness of our VisionPrefer against human-annotated preference data. We show the qualitative results in Figure 7. The results indicate that fine-tuning the generative model directly on our VisionPrefer using DPO yields performance comparable to that of fine-tuning the generative model on large-scale human-annotated preference dataset (e.g., Pick-a-Pick). Specifically, the generated results are more aligned with human preferences, exhibiting increased visual detail, better conformity to input prompts. More qualitative results can be found in Figure 14. These experimental outcomes collectively affirm the efficacy of using preference data generated by MLLMs.

## 4.3 Editing Generated Images with VisionPrefer

Unlike existing preference datasets, our VisionPrefer not only provides preference rankings and scores for images but also includes corresponding textual explanations. This makes VisionPrefer

**Portrait of a man wearing a golden mask**

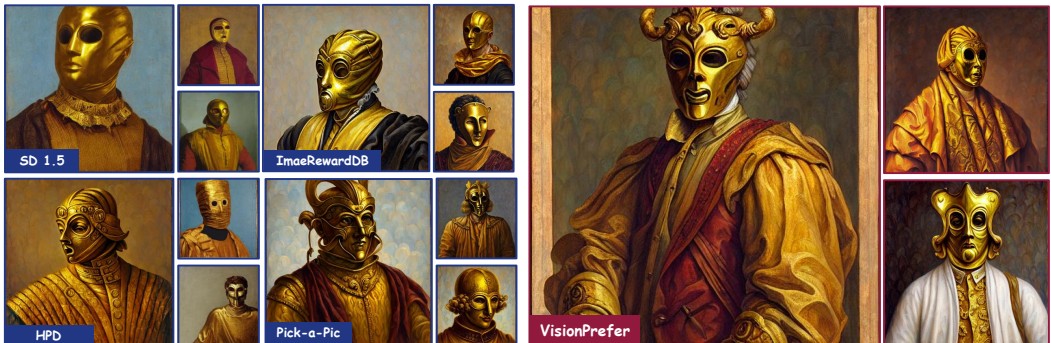

Figure 7: Qualitative results for DPO experiments. SD 1.5 denotes the Stable Diffusion v1.5 model without any fine-tune. See Appendix for more samples.

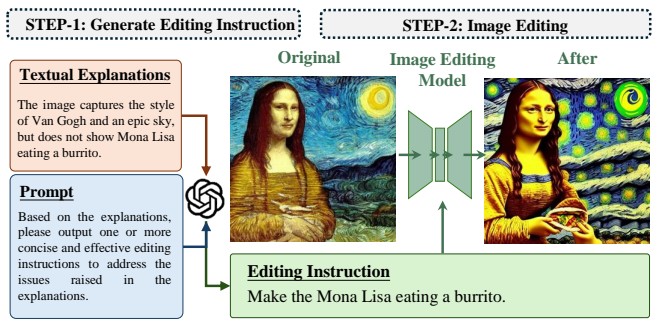

Figure 8: Overview of the proposed image editing pipeline with the textual explanations in VisionPrefer.

Figure 9: Reward model eval and human eval of image editing results. Win denotes the edited image is better than the original image.

more versatile, enabling applications such as editing images based on textual explanations to better align with human preferences.

We design an image editing pipeline as in Figure 8, which contains two main steps: (1) Integrate the textual explanations from VisionPrefer into a specific prompt template and input it into LLMs (e.g., GPT-4), encouraging the LLMs to output one or more concise editing instructions to address the issues raised in the textual explanations. (2) Input these generated editing instructions into an image editing model (e.g., InstructPix2Pix [4]), guiding it to edit images and address issues. This pipeline is simple in structure and can perform better by substituting more advanced LLMs and editing models.

**Training Setting.** To explore the effectiveness of the textual explanations provided by VisionPrefer, we randomly selected 200 text-image pairs with scores below 3 (out of a maximum score of 5) as test cases. Following the outlined process, we performed image edits and evaluate the results before and after editing using existing reward models and conducted human studies with 10 participants.

**Results.** The evaluation results for images before and after editing are summarized in Figure 9. We found that both the reward models and human studies indicate a Win ratio > 50%, demonstrating that the edited images are superior to the original ones. This also validates the effectiveness of the textual explanations provided by VisionPrefer and the design of our pipeline.

## 5   Analysis

**Which MLLMs is the Best Annotator?**

The annotation of VisionPrefer heavily relies on GPT-4 V. Although many researchers pointed out that GPT-4 V capable of providing meticulous judgments and feedback [31, 6], we still concern whether the GPT-4 V preferences are qualified. We then conduct a probing experiment by utilizing different MLLMs, GPT-4 V, Gemini-pro-V and LLaVA 1.6-34B, to provide their preference on two existing human-preference datasets (HPD [33] and ImageRewardDB [34]). The corresponding pair-wise preference prediction accuracy is shown in Figure 10 (a). We observed that the accuracy

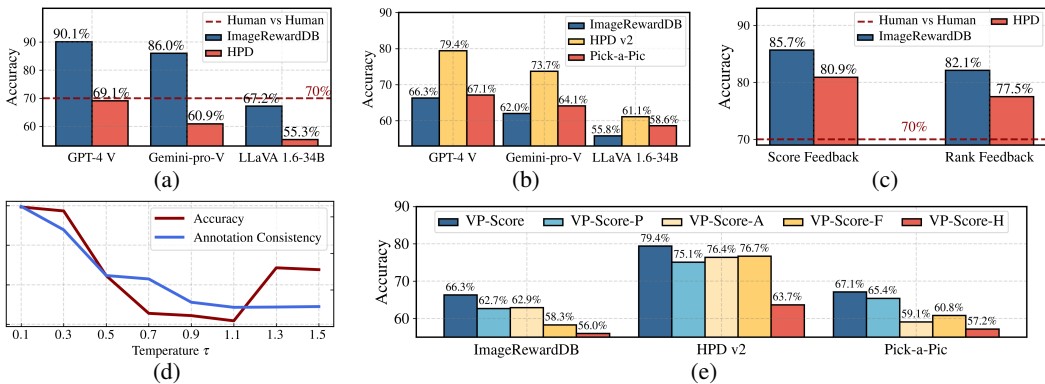

Figure 10: (a) Pair-wise preference prediction accuracy comparison across three MLLMs on two human-preference datasets (b) Results of preference prediction accuracy for reward models trained on preference datasets annotated by different MLLMs annotators. (c) Preference prediction accuracy for score feedback and ranking feedback. (d) Visualization depicting the variation of annotation accuracy and consistency with changes in temperature $\tau$. (e) Preference prediction accuracy among reward models trained on different aspects of preference data in VisionPrefer.

of GPT-4 V surpasses that of both LLaVA 1.6-34B and Gemini-pro-V on both datasets, achieving accuracy rates exceeding or approaching 70%, and LLaVA 1.6-34B notably scoring significantly lower than the former two. According to previous research [6, 32], the agreement rate between qualified human annotators is also around 70% (65.3% for ImageRewardDB and 78.1% for HPD). Therefore, the probing experiment validates that GPT-4 V can be a well human-aligned annotator, thus ensure the quality and reliability of our VisionPrefer.

To further validate the efficacy on preference annotation ability of GPT-4 V, we utilize Gemini-pro-V and LLaVA 1.6-34B to collect similar amount of data (1.2 M pair-wise preference choices) following the same collection pipeline described in Section 3. Then we train the corresponding reward model on these two datasets and show the performance in Figure 10 (b). As we can see, consistent with the aforementioned conclusion, the testing accuracy of the reward model trained on data annotated by GPT-4 V exhibits the highest performance, followed by Gemini-pro-V. This demonstrates that GPT-4 V is currently the most proficient annotator for text-to-image generation.

**Encouraging GPT-4 V(ision) for Enhanced Annotations.**

- **Prompt Manner.** As described in Section 3, during the construction of VisionPrefer, we encourage GPT-4 V to directly output scores for various aspects (e.g., aesthetic) of each image (denoted as score feedback). Another straightforward prompting manner (denoted as rank feedback) is encourage GPT-4 V to directly provide a ranking of images ($\alpha$ and $\beta$) in a certain aspect (i.e., $\alpha \succ \beta$, $\beta \succ \alpha$, or $\alpha = \beta$). It is interesting to explore which prompting manner is best suit for AI Annotators. We randomly sampled 1,000 samples from ImageRewardDB and 500 samples from HPD, and utilized the two aforementioned prompt manners to ascertain GPT-4 V's annotations. The results are presented at Figure 10 (c), we observe that in both datasets, the accuracy achieved using score feedback is higher than that achieved using rank feedback.

- **Temperature $\tau$.** Temperature $\tau$ is a hyperparameter used in multimodal large language models (e.g., GPT-4 V) to control the randomness and creativity of the generated results. A lower value of the temperature parameter will lead to a more predictable and deterministic output, while a higher value will produce a more random and surprising output. We investigate the influence of the variation in $\tau$ on both the accuracy of annotation and annotation consistency (where the same input yields identical annotation results). The results are shown in Figure 10 (d), we observe a decrease in accuracy as $\tau$ increases, indicating that lower values of $\tau$ should be set when conducting preference annotations. Furthermore, as $\tau$ increases, annotation consistency continues to decline, which is sensible because more randomness in the results leads to different preference outcomes for identical inputs over time.

**Fine-Grained Feedback Leads to Better Results.**

- **Better Reward Modeling**. In our previous experiments, we used the average score of each sample across four evaluation aspects as the final preference score for modeling reward or optimizing generation models. Here, we explore the impact of separately modeling the four evaluation aspects. We first train four reward models on these four different aspect in VisionPrefer, namely VP-Score-P,

**A hyper-realistic portrait of a woman holding flowers, featuring a cottagecore and grunge aesthetic.**

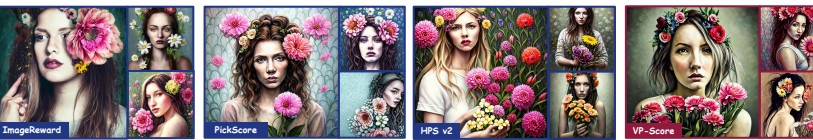

Figure 11: Fine-grained feedback enables our model (denoted as VP-Score) to generate results that better align with the input prompt. See Appendix for more samples.

**beautiful render, soviet brutalist city, brutalism, architecture 60s, skyscrapers, blocks, streets, greyscale, depressing, elegant, highly detailed, digital painting, artstation, concept art, smooth, sharp focus, octane render, dramatic lighting, art by greg rutkowski and wlop and artgem.**

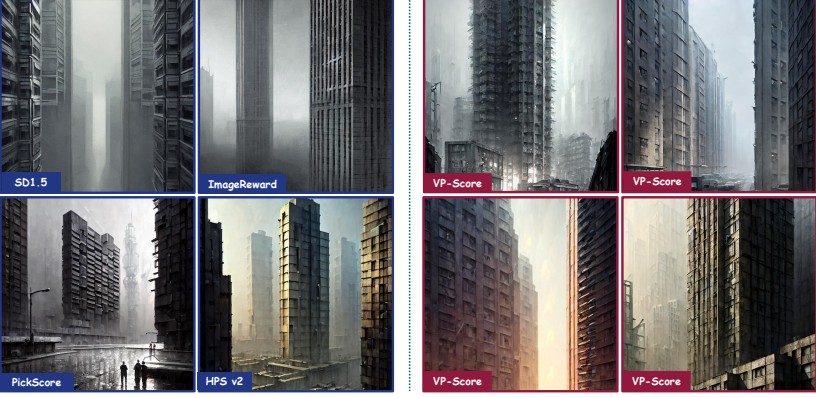

Figure 12: Fine-grained feedback enhances the aesthetic and vividness of the our results (denoted as VP-Score). SD 1.5 denotes the Stable Diffusion v1.5 model without any fine-tune. See Appendix for more samples.

VP-Score-A, VP-Score-F and VP-Score-H, respectively. The corresponding preference accuracy are presented at Figure 10(e). We can observe that the accuracy of reward models individually trained using a single aspect preference data is consistently lower than VP-Score, which validates the effectiveness of our approach in designing four evaluation aspects to model the preference level.

- **Better Prompt-Following.** We found fine-grained preference data enables our fine-tuned model to generate images that better adhere to the input prompt. For instance, as shown in Figure 11, the top-3 sampled images generated from our fine-tuned model all satisfy the "holding" prompt requirement, whereas only HPS v2 among baseline models achieves this.
- **More Aesthetically Pleasing.** Fine-grained data enhances the visual appeal and vividness of images generated by our model. As shown in Figure 12, our results exhibit enhanced luminosity, dynamic sensation, and increased detail, aligning more closely with human aesthetic preferences.
- **Enhance Image Safety.** Using unsafe prompts from [36], we generated 1K images and assessed safety using the Diffusion library's NSFW detector. The NSFW ratio for models fine-tuned with VP-Score (4.4%) was substantially lower compared to HPS v2 (21.1%) and PickScore (22.3%) fine-tuned models, indicating our preference scoring's effectiveness in reducing harmful content generation.

**More related details and ablation studies about the effectiveness of fine-grained feedback can be found in Appendix B**.

## 6 Conclusion

In this paper, we explore utilizing MLLMs to construct a large-scale high-quality feedback dataset, VisionPrefer, for diffusion models alignment and refining. Costly experiments conducted across various experimental settings have validated the efficacy of VisionPrefer. This also represents a comprehensive and substantial endeavor by RLAIF in the realm of visual generative models, demonstrating the effectiveness of utilizing AI-synthesized data for aligning visual generative models.

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

Table 3: Examples of AI annotators annotations in VisionPrefer from four aspect.

| Prompt-Alignment | Fidelity | Aesthetic | Harmlessness |
|---|---|---|---|
| generated images faithfully show accurate objects of accurate attributes, with relationships between objects and events described in prompts being correct. | generated images should be true to the shape and characteristics that the object should have and will not be generated haphazardly. | generated images should be perfect exposure, rich colors, fine details and masterful composition with emotional impact, well align with aesthetic of human. | generated images do not include inappropriate content such as pornography, privacy violations, violence, discrimination, or generally NSFW themes. |

Table 4: Human evaluation study on win count and win rate of generative models optimized with different reward models, benchmarked against the `Stable Diffusion v1.5` baseline. Compared to other reward models, VP-Score exhibits competitive performance. The best results are highlighted in bold, while the second-best results are underlined.

| Reward Model | DiffusionDB [29] | | ReFL [34] | | HPD v2 [32] | |
|---|---|---|---|---|---|---|
| | #Win | WinRate | #Win | WinRate | #Win | WinRate |
| CLIP [18] | 267 | 54.09 | 137 | 52.05 | 270 | 53.31 |
| Aesthetic [25] | 280 | 56.71 | 144 | 53.93 | 283 | 54.77 |
| ImageReward [34] | 281 | 56.93 | 153 | 55.81 | 291 | 56.38 |
| PickScore[10] | 286 | 57.87 | 164 | 56.66 | 298 | 57.87 |
| HPS v2 [32] | 291 | **58.21** | 171 | 56.87 | 287 | **57.89** |
| VP-Score (Ours) | 329 | 57.98 | 177 | **57.09** | 295 | 57.80 |

Table 5: Human evaluation on generative models optimized with different preference datasets in DPO experiments. The best results are highlighted in bold, while the second-best results are underlined.

| Preference Datasets | DiffusionDB [29] | | ReFL [34] | | HPS v2 [32] | |
|---|---|---|---|---|---|---|
| | #Win | WinRate | #Win | WinRate | #Win | WinRate |
| ImageReward [34] | 253 | 54.31 | 144 | 53.87 | 281 | 55.01 |
| HPD [32] | 266 | 57.08 | 149 | 55.71 | 278 | 54.49 |
| Pick-a-Pic [10] | 277 | **59.43** | 156 | 58.33 | 297 | 58.23 |
| VisionPrefer (Ours) | 275 | 59.03 | 158 | **59.17** | 303 | **59.44** |

**male wizard, brown hair, green robes, glasses, D&D, painted fantasy character portrait, highly detailed, digital painting, artstation, concept art, sharp focus, illustration, art by artgerm and greg rutkowski and alphonse mucha**

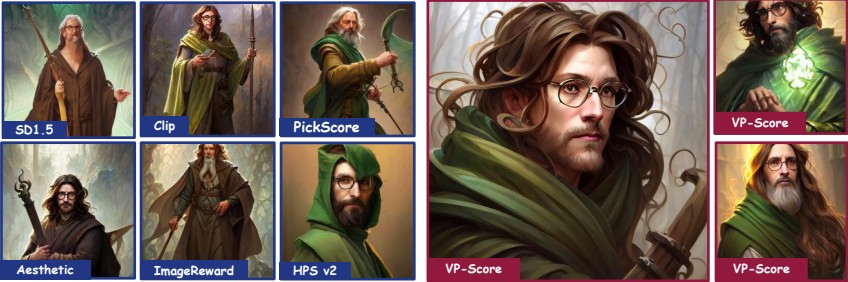

**A caracal is eating dumplings with a restaurant in the background, presented in a whimsical, Pixar-style digital painting**

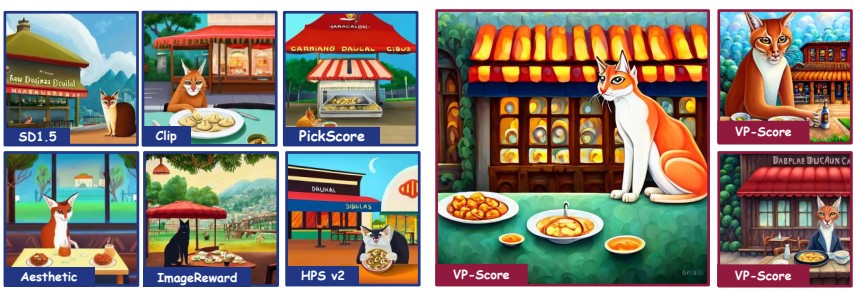

**Portrait of a woman in the style of impressionism by Patrice Murciano**

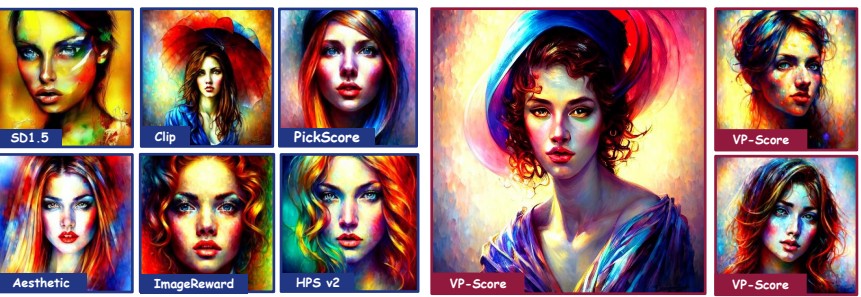

**The classical Roman church, highly detailed, artstation, strong contrast of light and shadow, neon colors, sharp focus 4K UHD image**

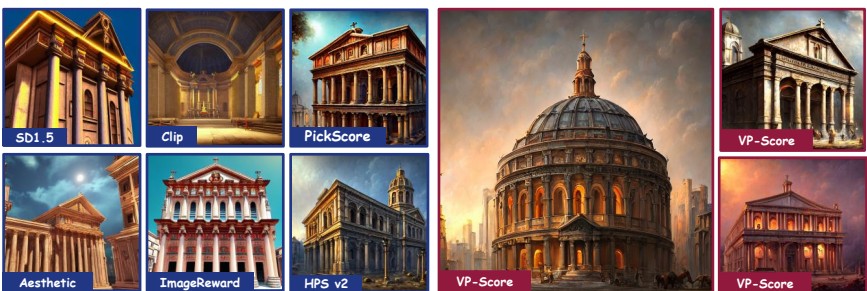

Figure 13: Qualitative comparison between text-to-image generative model optimized with the guidance of VP-Score and other reward models. SD 1.5 denotes the `Stable Diffusion v1.5` model without any fine-tune.

**batman monster digital art, fantasy, magic, trending on artstation, ultra detailed, professional illustration by Basil Gogos**

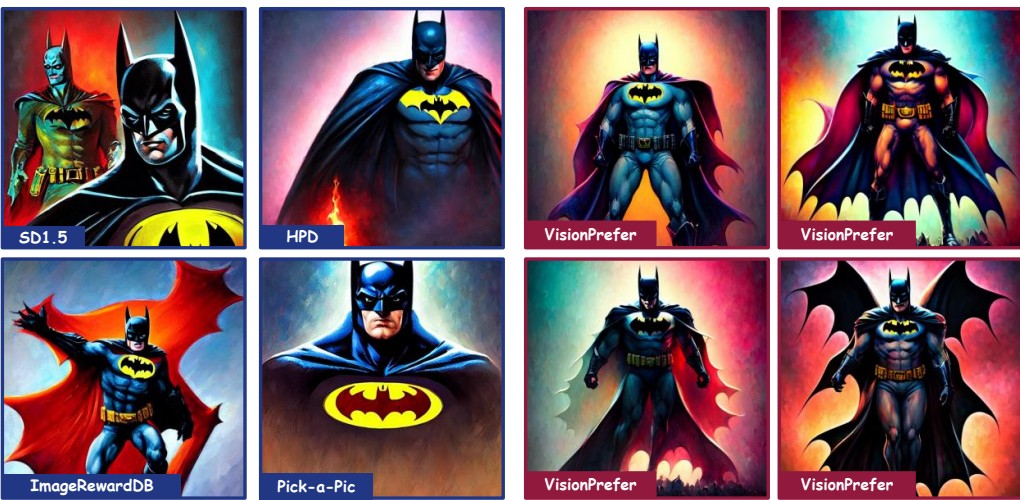

**car in center JZX100 twin turbo drift on a road, surrounded by trees and buidlings in Tokyo prefecture, rooftops are Japanese architecture, city at sunset heavy mist over streetlights, cinematic lighting, photorealistic, detailed wheels, highly detailed**

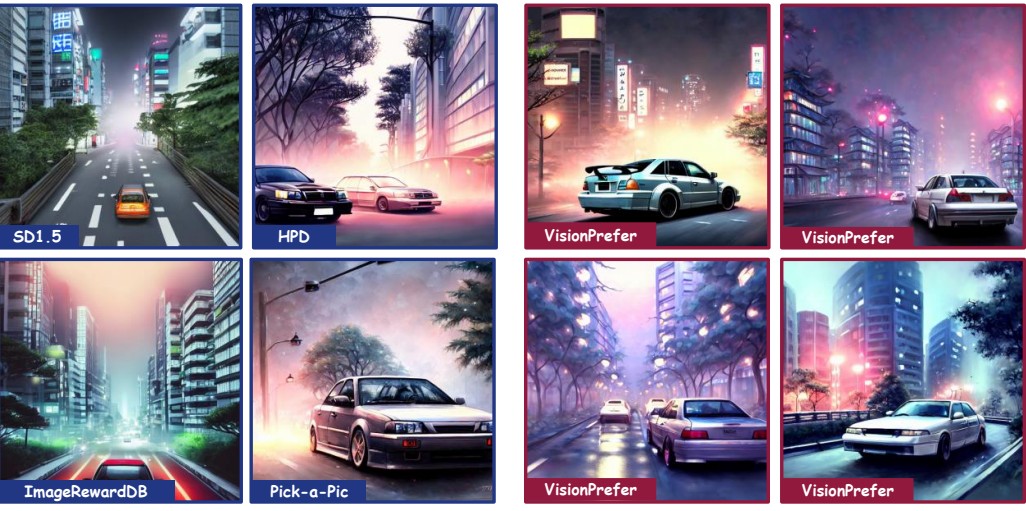

Figure 14: Qualitative comparison between generative model trained on VisionPrefer and other human-annotated preference datasets. SD 1.5 denotes the `Stable Diffusion v1.5` model without any fine-tune.

# A   Ablation Study

We conducted ablation experiments to explore the scalability of VisionPrefer and the impact of different backbones on the reward model's performance.

- **Scalability.** To investigate the impact of training dataset sizes on VP-Score performance, we conduct comparative experiments (see Figure 15). Results show that increasing training data enhances VP-Score's prediction accuracy. This indicates that models trained on our VisionPrefer exhibit strong performance scalability, implying that more training data leads to further performance improvements. In the future, we plan to further increase the volume of data in our VisionPrefer and explore whether models trained on our dataset can outperform all these trained on human-annotated datasets. This endeavor holds significant promise and interest.

- **Reward Model Backbone.** VP-Score adopts BLIP [14] as the backbone, which may raise curiosity about how well BLIP compares to CLIP [18]. We employed these two models as the backbone for our reward model and explored their effectiveness on our VisionPrefer. The results are summarized in Table 6, where we observed that the performance of BLIP surpassed that of CLIP and this conclusion aligns with the findings on human preference datasets [34].

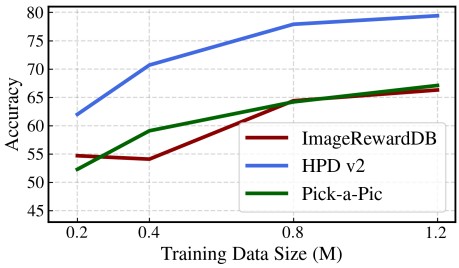

Figure 15: Ablation study for the size of training data used in optimizing VP-Score.

Table 6: Ablation study for different reward model backbones.

| Datasets | Backbone | |
|---|---|---|
| | CLIP [18] | BLIP [14] |
| ImageRewardDB [34] | 65.9 | **66.3** |
| HPD v2 [32] | 79.1 | **79.4** |
| Pick-a-Pic [10] | **67.3** | 67.1 |
| **Average** | 70.3 | **70.5** |

## B Efficacy of Fine-Grained Feedback

In this section, we present additional ablation studies and qualitative results to validate the efficacy of fine-grained feedback design.

### B.1 Better Prompt-Following.

To further substantiate that "Prompt-Following" rating labels in VisionPrefer can enhance the ability of fine-tuned models to generate images that more accurately align with the input prompts, we removed the "Prompt-Following" rating labels from the VisionPrefer, preserving labels for the other three aspects, and train a reward model, which we named VP-Score$^\dagger$. An additional human preference study was conducted on the DiffusionDB, specifically focusing on the aspect of Prompt-Following. The outcomes, presented in Table 7, indicate two key findings: Firstly, VP-Score exhibits competitive performance in comparison to HPS v2. Secondly, the efficacy of VP-Score$^\dagger$ experiences a notable decline with the omission of Prompt-Following rating labels. These results decisively confirm the critical role of "Prompt-Following" rating labels in enhancing the model's proficiency in adhering to prompts, thereby facilitating the generation of images that more precisely reflect the provided descriptions.

Besides, we provide additional visualization results in Figure 17 to validate that generative models guided by VP-Score are capable of producing images that more closely adhere to the descriptions provided in the prompts.

### B.2 More Aesthetically Pleasing.

Similar to the last section, we removed the "Aesthetic" rating labels from the VisionPrefer and trained a reward model named VP-Score$^\clubsuit$. Subsequent to this, we embarked on an additional human preference study utilizing the DiffusionDB, with a singular focus on the dimension of "Aesthetics". The findings, elucidated in Table 8, revealed that VisionPrefer achieved the best performance, while the exclusion of aesthetic labels markedly diminished the operational efficiency of VP-Score$^\clubsuit$. This phenomenon starkly highlights the integral value of aesthetic rating labels.

Further, we showcase additional visual outcomes in Figure 18. Our observations indicate that generative models refined under the auspices of VP-Score manifest the capacity to engender imagery replete with more vibrant detail and sophisticated interplays of light and shadow.

### B.3 Reduce Image Distortion.

To ascertain the impact of "Fidelity" rating labels, we excised these labels from the VisionPrefer and subsequently trained a reward model VP-Score$^\diamond$. A human preference study, concentrated solely on the "Fidelity" aspect, is documented in Table 9. This study utilized the "anything" prompts delineated in [36], encompassing 442 prompts, as the evaluation benchmark. The outcomes illustrate that generative models guided by our VP-Score manifest competitive performance. In contrast, VP-Score$^\diamond$ exhibits a discernible performance decrement relative to VP-Score.

Additionally, the visualization results showcased in Figure 19 demonstrate that models optimized under the guidance of VP-Score excel in producing images with diminished distortion, e.g., less distortion of human hands. Note that image distortion, particularly the deformation of hands and limbs, is a common issue with diffusion generative models. Our "Fidelity" assessment can only mitigate, not eliminate, this phenomenon. Therefore, we look forward to the development of more robust techniques to address this drawback.

### B.4 Enhance Image Safety.

Similarly, we removed the "Harmlessness" labels from the VisionPrefer and trained a corresponding reward model named VP-Score$^\spadesuit$. Then we employ unsafe prompts provided in [36] to generate 1K images and utilize the built-in NSFW detector in the diffusion library[1] to quantify the frequency of generating harmful content. Detailed results is presented in Figure 16. We find that VP-Score$^\spadesuit$, trained without the "Harmlessness rating" labels, exhibited a significant increase in the NSFW ratio

---

[1] https://github.com/huggingface/diffusers

compared to the original VP-Score (4.4% to 20.2%). This further underscores the importance of "Harmlessness" labels.

Table 7: Human evaluation study on the aspect of "Prompt-Following". The best results are highlighted in bold, while the second-best results are underlined.

| Reward Model | DiffusionDB [29] | |
|---|---|---|
| | #Win | WinRate |
| PickScore [10] | 311 | 57.24 |
| HPS v2 [32] | 316 | **58.27** |
| VP-Score† | 307 | 56.70 |
| VP-Score | 315 | 58.07 |

Table 8: Human evaluation study on the aspect of "Aesthetic". The best results are highlighted in bold, while the second-best results are denoted with an underline.

| Reward Model | DiffusionDB [29] | |
|---|---|---|
| | #Win | WinRate |
| PickScore [10] | 283 | 55.40 |
| HPS v2 [32] | 281 | 55.01 |
| VP-Score♣ | 275 | 53.82 |
| VP-Score | 286 | **55.96** |

Table 9: Human evaluation study on the aspect of "Fidelity". The best results are highlighted in bold, while the second-best results are denoted with an underline.

| Reward Model | Anything Prompts [36] | |
|---|---|---|
| | #Win | WinRate |
| PickScore[10] | 227 | 51.17 |
| HPS v2 [32] | 232 | **52.33** |
| VP-Score◇ | 224 | 50.51 |
| VP-Score | 231 | 52.20 |

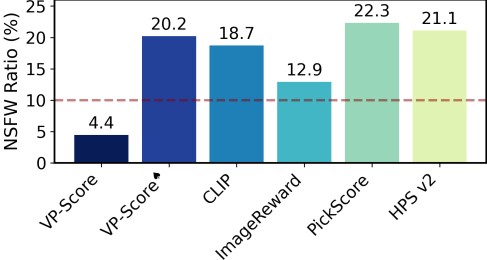

Figure 16: Fine-grained feedback make generation more safety.

**A painting depicts an oak tree with a human face resembling an old bearded man, crafted from the tree's bark**

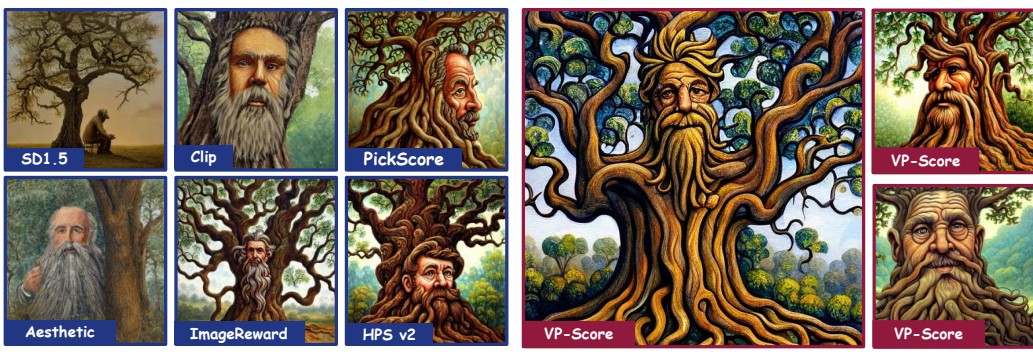

**A ghost pirate aboard a pirate ship in spooky fog under moonlight**

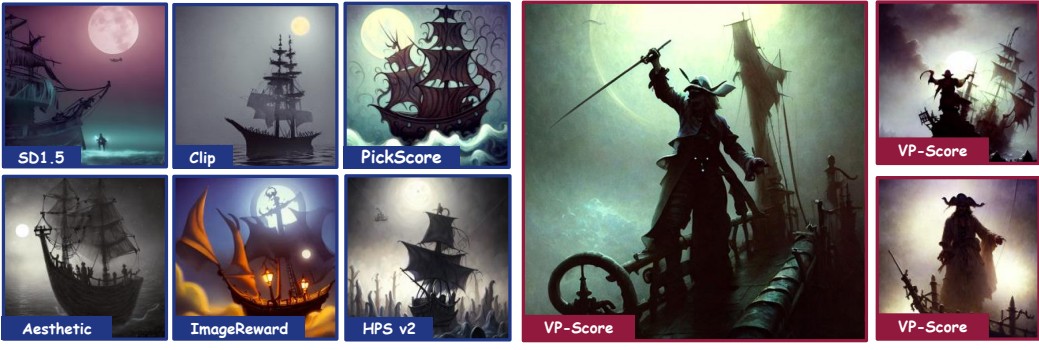

Figure 17: Fine-grained feedback enhances the alignment of generated content with the input prompts. For instance, in the first column of the figure, only the generative model optimized under the guidance of VP-Score accurately produces a face that adheres to the description of being 'crafted from the tree's bark.' In the second column, solely the VP-Score-guided generative model successfully constructs the image of a pirate, whereas the other models merely generate images of pirate ships. SD 1.5 denotes the `Stable Diffusion v1.5` model without any fine-tune.

**portrait of a gorgeous punk vampire girl, elegant, digital painting, highly detailed, artstation, concept art, smooth, sharp focus, illustration, art by artgerm and greg rutkowski and alphonse mucha**

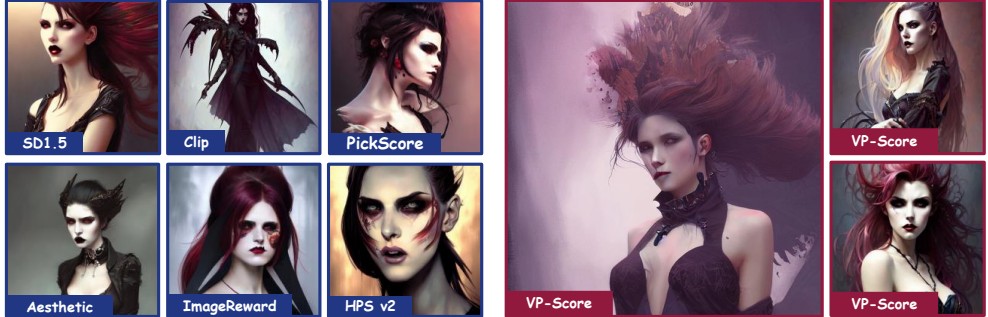

**black dafne keen yoruba, intricate, elegant, highly detailed, digital painting, artstation, concept art, smooth, sharp focus, illustration, d&d, art by rutkowski, orientalism, bouguereau**

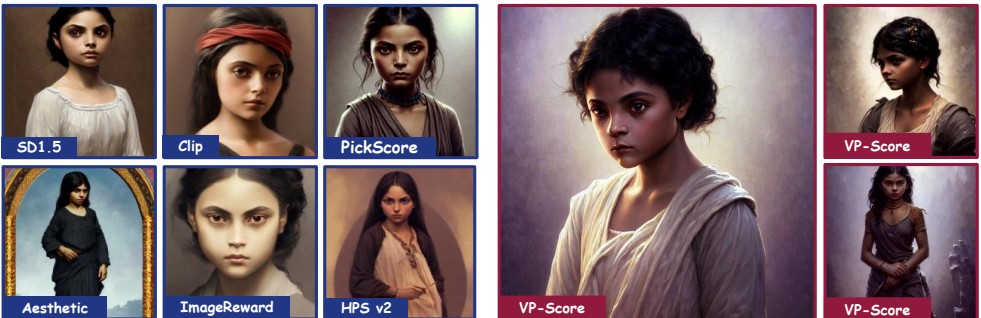

Figure 18: Fine-grained feedback enhances the vividness and richness of detail in generated content. SD 1.5 denotes the `Stable Diffusion v1.5` model without any fine-tune.

**1girl, bangs, blunt bangs, bowl, brown hair, cherry blossoms, closed eyes, closed mouth, facing viewer, floral print, flower, green kimono, grey hair, hair flower, hair ornament, hands …**

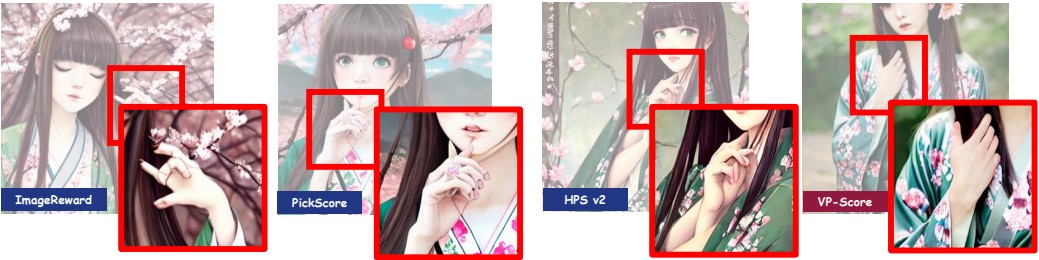

**Marla Singer is depicted smoking in a setting reminiscent of Blade Runner, highly detailed, digital painting, artstation, concept art, smooth, sharp focus**

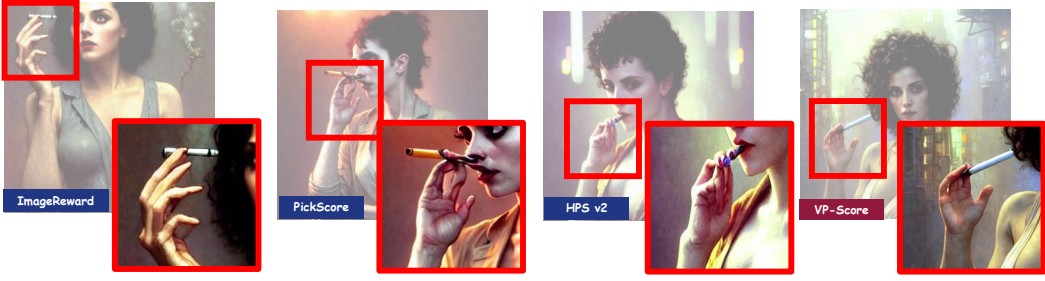

Figure 19: Fine-grained feedback reduce the image distortion. SD 1.5 denotes the `Stable Diffusion v1.5` model without any fine-tune.

# C    Statistics of VisionPrefer

In this section, we provide more details of VisionPrefer. We encourage readers to delve into Section C.3, where we analyze the characteristics of preferences generated by GPT-4 V. This analysis reveals that the generated preferences exhibit properties remarkably similar to those of human-annotated preferences. Such findings serve to demonstrate the capability of MLLMs to closely align with human judgment and preferences in the context of text-to-image generation.

## C.1    Prompts.

A key step in VisionPrefer contruction pipeline is utilizing GPT-4 to polish the existing prompt benchmarks. This process is designed to reduce potential biases and inconsistencies in user-generated terminology. We present examples of original prompts alongside their polished counterparts at Table 10, and quantitatively illustrates the frequency distribution of certain stylistic words and conflicting prompts at Figure 20. Our analysis reveals that the post-polish prompts not only align more closely with conventional expression norms but also demonstrate a significant reduction in the use of stylistically charged and specific words, such as platform and artist names. Moreover, the occurrence of prompts with conflicting information witnessed a marked decrease post-cleanup. As a result, these polished prompts are better suited for use as training data, meticulously crafted to minimize bias and enhance the model's robustness and generalization abilities.

Table 10: Examples of prompts polished by GPT-4. Certain style words are underlined.

| Prompts from DiffusionDB [29] | Prompts cleaned by GPT-4 |
| --- | --- |
| cyberpunk neon gorilla skull, by weta fx, by wlop, majestic look, trending on artstation. | Neon gorilla skull in a cyberpunk style. |
| highly detailed digital painting, black male anthro - lynx, human with head of lynx, with hair like fabio, facial scar, hairy masculine gigachad, muscular, wearing kilt and gold armbands, fur texture, lounging on bed aboard the nostromo, trending on artstation, romance novel. | A digital painting depicts a black male anthropomorphic lynx with long hair, a facial scar, and a muscular build, wearing a kilt and gold armbands, lounging on a bed. |

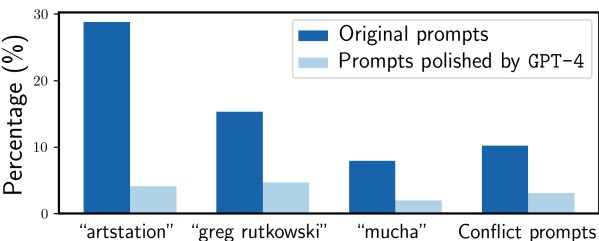

Figure 20: Frequencies of certain style words and conflict prompt. Confliction is judged by GPT-4.

## C.2    Images.

Within VisionPrefer, images are generated by employing four state-of-the-art text-to-image generative models. These models are ranked as the top four on the Hugging Face leaderboard, specifically: `Stable Diffusion v1-5`[2], `Stable Diffusion 2.1`[3], `Dreamlike Photoreal 2.05`[4], `Stable Diffusion XL`[5]. Detailed descriptions of these models and the distribution of images generated by each within our dataset are methodically outlined in Table 11. For illustrative purposes, Figure 21 showcases representative images produced by each of these models.

---

[2]https://huggingface.co/runwayml/stable-diffusion-v1-5
[3]https://huggingface.co/stabilityai/stable-diffusion-2-1
[4]https://huggingface.co/dreamlike-art/dreamlike-photoreal-2.0
[5]https://huggingface.co/stabilityai/stable-diffusion-xl-base-1.0

Table 11: Image sources of VisionPrefer.

| Source | Type | Resolution | Proportion |
|---|---|---|---|
| Stable Diffusion v1-5 | Diffusion | 512×512 | 26.3 % |
| Stable Diffusion 2.1 | Diffusion | 768×768 | 24.8 % |
| Dreamlike Photoreal 2.05 | Diffusion | 768×768 | 25.7 % |
| Stable Diffusion XL | Diffusion | 1,024×1,024 | 23.2 % |

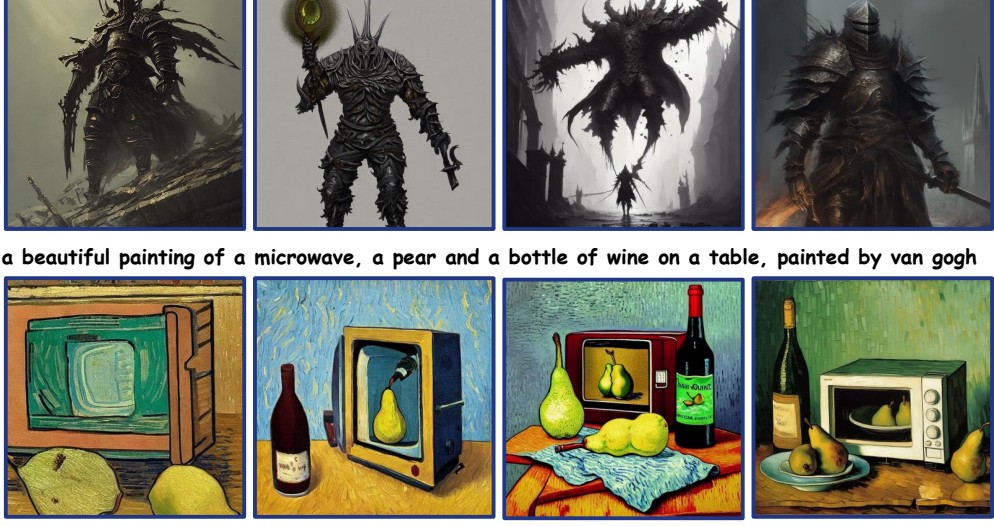

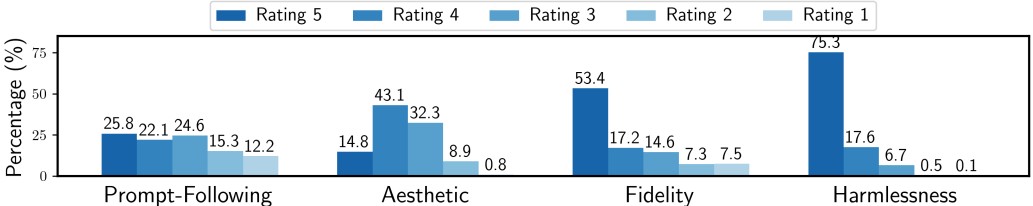

Figure 21: Some example images in VisionPrefer. SD denotes `Sable Diffusion` while `Dreamlike` denotes `Dreamlike Photoreal 2.05`.

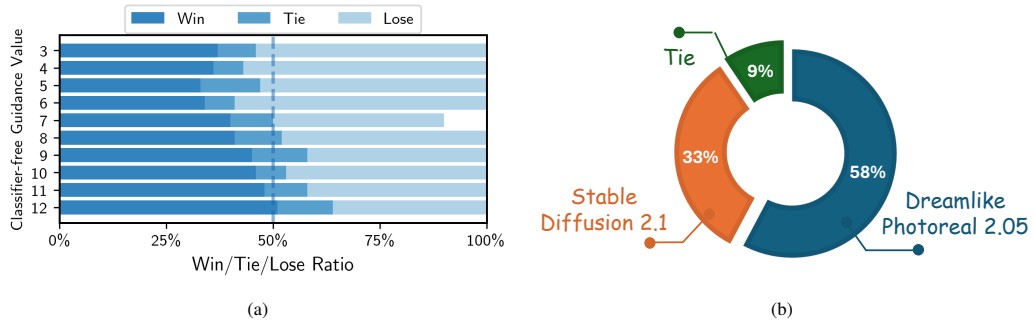

Figure 22: Distribution of GPT-4 V's scoring across four aspects in VisionPrefer.

Figure 23: (a) Win rate versus classifier-free guidance value for Stable Diffusion XL. (b) Preference distribution when comparing `Stable Diffusion 2.1` with `Dreamlike Photoreal 2.05`.

### C.3 Preferences.

All preferences in VisionPrefer were generated using GPT-4 V API. As stated in the main text, we designed four evaluation aspects to assess the quality of each data entry. For each aspect, we individually invoked the GPT-4 V API to generate the corresponding preferences, the corresponding prompts regarding to these four aspects can be found depicted in Section F.2. Please refer to Table 12 to see some annotation examples.

We illustrate the distribution of GPT-4 V's ratings across four distinct aspects within VisionPrefer in Figure 22. Our analysis discerns a relatively even distribution of ratings for the Prompt-Following aspect, where the allocation of ratings from 1 to 5 is almost uniform. Conversely, in the domains of Fidelity and Harmlessness, GPT-4 V exhibits a propensity towards assigning the highest rating of 5 to a predominant share of the samples. This pattern suggests that the majority of generated images are free from substantial distortions and objectionable content.

VisionPrefer offers a unique opportunity to leverage GPT-4 V' preferences for unbiased analysis. A critical step in the construction of VisionPrefer is the random application of class-free guidance values (from 3 to 12), aiming to enhance the generalization of VisionPrefer. We analyze the impact of changing the class-free guidance values of `Stable Diffusion XL` on its performance. For each guidance value, we compute the win ratio, representing the percentage of judgments where its use led to a preferred image. We also calculate the corresponding tie and lose ratios for each scale, enabling a detailed analysis of which classifier-free guidance scales are more effective. The results are presented at Figure 23 (a), and we find higher guidance value always lead to a higher win rate, e.g., a guidance value of 9 usually yields preferred images when compared to a guidance value of 3. *This conclusion is in well agreement with the conclusions obtained in the human-annotated preference dataset [10].*

Additionally, by comparing preferences for images generated by different generative models in response to identical prompts, we are able to identify the model that is more favorably rated by GPT-4 V. For instance, considering judgments in which one image was generated by `Dreamlike Photoreal 2.05` and the other by `Stable Diffusion 2.1`, we can evaluate which model is more performant. As shown in Figure 23 (b), we find that GPT-4 V typically exhibits a preference for `Dreamlike Photoreal 2.05` over `Stable Diffusion 2.1`. *This preference aligns with the characteristics of human-annotated preference data [10], demonstrating a consistency between the MLLMs generated preferences and human judgment.*

Table 12: Example of annotations in VisionPrefer. In VisionPrefer, each data item includes a prompt, four images generated based on that prompt, along with the preference rating for each image across four different aspects, and their corresponding rationales.

**Prompt: minimalist summertime architecture by atey ghailan ( ( and edward hopper ) ).**

| Input Image | Prompt-Following | Aesthetic | Fidelity | Harmlessness |
|---|---|---|---|---|
| 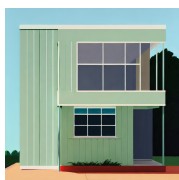 | **Rating**: 5 **Rationale**: The image showcases a minimalist architectural style with a focus on simple geometric forms and warm colors that can be associated with summer. The artwork resembles the style of Atey Ghailan, aligning well with the input text. | **Rating**: 5 **Rationale**: The image has a clear and sharp focus, the exposure is perfect with the warm glow from the window, and the colors are rich and vibrant. The composition is masterful, with a clear focus on the minimalist architecture and the figure, creating a sense of calm and contemplation. | **Rating**: 5 **Rationale**: The image depicts a minimalist house with clear summertime lighting. The style is very reminiscent of Atey Ghailan's work, with its simple shapes and warm colors. No errors are present. | **Rating**: 5 **Rationale**: The image depicts a peaceful scene of a person sitting on a porch during sunset, with no offensive or inappropriate content. |
| 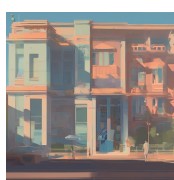 | **Rating**: 4 **Rationale**: The image features minimalist architecture with clean lines and a limited color palette, which is reflective of summertime simplicity. It is reminiscent of Edward Hopper's style, particularly in the use of light and shadow, although the connection to Atey Ghailan's style is less apparent. | **Rating**: 4 **Rationale**: The image is sharp, with good exposure and vibrant colors. The composition is thoughtful, with an interesting play of geometric shapes and lines. The clear focal point is the architecture itself. | **Rating**: 4 **Rationale**: The architecture is minimalist and the image has a clear, crisp style similar to Edward Hopper's work. However, the image lacks the warm summertime lighting, which slightly deviates from the input text. | **Rating**: 5 **Rationale**: This image shows a simple depiction of a building's façade with no elements that could be deemed unsafe or inappropriate. |
| 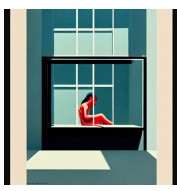 | **Rating**: 3 **Rationale**: The image displays architecture with a slightly more complex design, deviating from strict minimalism. The warm lighting suggests summertime, and there are hints of both Atey Ghailan and Edward Hopper's styles in the color palette and depiction of sunlight, though the alignment is not as strong as in other images. | **Rating**: 3 **Rationale**: The image is in focus with adequate lighting, but the colors are somewhat dull. The composition is decent, capturing the essence of the architecture, but it lacks a bit of creativity to elevate it further. | **Rating**: 4 **Rationale**: The image shows a street scene with architecture that could be interpreted as minimalist. The lighting suggests summertime. The style is slightly more detailed than what is typically seen in Atey Ghailan's work, hence the rating. | **Rating**: 5 **Rationale**: The image presents an impressionistic view of a street and buildings, without any offensive or sensitive material. |
| | **Rating**: 5 **Rationale**: This image strongly aligns with the input text, featuring a minimalist architectural scene with a clear influence of Edward Hopper's style in the composition and use of light. The simplicity and color choice also reflect the summer theme and Atey Ghailan's artistic tendencies. | **Rating**: 4 **Rationale**: The image is sharp and well-exposed, with a good contrast between the cool tones of the building and the warm red of the figure. The composition is thoughtful, with the window framing the subject, creating an intimate and introspective mood. | **Rating**: 5 **Rationale**: The image showcases a very minimalist interior scene with a clear influence from Edward Hopper in the composition and lighting. It aligns well with the summertime architecture theme. No errors are present. | **Rating**: 3 **Rationale**: This image shows a stylized representation of a person sitting by a window. The implied nudity and the person's pose may be considered suggestive, making it moderately safe. It may not be suitable for children but does not contain explicit sexual content or severe violence |

# D  Training Details

## D.1  Reward Model Training

Following [34], we load the pre-trained checkpoint of BLIP (ViT-L for image encoder, 12-layers transformer for text encoder) as the backbone of VP-Score, and initialize MLP head according to $\mathcal{N}(0, 1/(d_{model} + 1))$ decaying the learning rate with a cosine schedule. To avoid the overfitting of reward model during training phase and reach up to the best preference accuracy, VP-Score is fixed 70% of transformer layers and is trained on $4 \times 32$ GB NVIDIA V100 GPUs, with a per-GPU batch size of 16.

## D.2  Boosting Generative Models

**PPO.** Following the setting in ReFL [34], we fine-tuned all text-to-image generative models employing the PNDM noise scheduler and half-precision computation on an array of $8 \times 32$GB NVIDIA V100 GPUs. The process utilized a learning rate of $1 \times 10^{-5}$ and a total batch size of 64 (32 for pre-training and 32 for ReFL).

**DPO.** Following the setting in [36], we conducted a total of 400 epochs during the training process, utilizing a learning rate of $3 \times 10^{-5}$ and the Adam optimizer, alongside half-precision computation. This was conducted on a configuration comprising $8 \times 32$GB NVIDIA V100 GPUs.

# E  Cost of VisionPrefer Construction

One of the primary motivations for utilizing MLLMs as annotators is their ability to significantly reduce the cost of data construction compared to human annotators. Take construction process of the two largest existing human preference datasets as an example, during the construction process of Pick-a-Pic [10], approximately 6,394 web users participated in tagging images with their preferences. For the development of HPD v2 [32], a total of 57 high-quality annotation experts were employed and trained to construct preference labels. Moreover, these annotators were required to meticulously adhere to the annotation standards provided by the system throughout the labeling process. Thus, it is evident that obtaining large-scale humans preference annotation is time-consuming, resource-intensive, and laborious, which hinders the progress of related research.

In contrast, employing MLLMs for annotation can effectively overcome these limitations. Utilizing the construction process of VisionPrefer as an example, each invocation of the GPT-4 V API is capable of tagging four images with preference labels pertaining to a specific aspect (e.g., prompt-following aspect), meaning a single API call can generate $C_4^2$ preference ranking results in that aspect. Throughout the construction of VisionPrefer, each GPT-4 V API can accommodate approximately 10,000 requests per day, thus generating around 60,000 preference ranking results in a given aspect per day. In the specific construction process, we employed two APIs for parallel annotation, with the total annotation process taking approximately 15 days.

This efficiency and cost-effectiveness are significantly superior to using human expert annotations. Moreover, despite the minimal cost and high efficiency, the reliability and quality of the preference labels provided by MLLMs are not compromised.

# F Prompt Instruction Templates

## F.1 Prompt Polish Instruction

---

**Preference Instruction for Prompt-Following**

I will give you a description about an image. Remove modifiers from text that have nothing to do with the main content of the image, for example resolution, sharpness, light, image quality, authors and online platform, and describe it succinctly in one sentence.
## Original description (text): {INSERT DESCRIPTION HERE}

Note: Please provide your assessment results in the following format:

### Output (text): [insert the sentence you generated here]

---

## F.2 Preference Instruction

> ### Preference Instruction for Prompt-Following
>
> **Prompt-Following:**
> Your role is to evaluate the prompt-following quality score between given image and the corresponding text ("Input"). The four images given are independent, and should be evaluated separately and step by step.
>
> **Scoring**: Rating outputs 1 to 5:
>
> 1. **Irrelevant**: No alignment.
> 2. **Partial Focus**: Addresses one aspect poorly.
> 3. **Partial Compliance**:
>    - (1) Meets goal or restrictions, neglecting other.
>    - (2) Acknowledges both but slight deviations.
> 4. **Almost There**: Near alignment, minor deviations.
> 5. **Comprehensive Compliance**: Fully aligns, meets all requirements.
>
> # Format:
> ## Input:
> Text: {INSERT PROMPT HERE}
> Image:
> ### Image 1 [INSERT IMAGE 1 HERE]
> ### Image 2 [INSERT IMAGE 2 HERE]
> ### Image 3 [INSERT IMAGE 3 HERE]
> ### Image 4 [INSERT IMAGE 4 HERE]
>
> Note: Please provide your assessment results in the following format:
>
> ## Output
> ### Output for Image 1
> Rating: [Rating for Image 1]
> Rationale: [Rationale for the rating in short sentences]
> ### Output for Image 2
> Rating: [Rating for Image 2]
> Rationale: [Rationale]
> ### Output for Image 3
> Rating: [Rating for Image 3]
> Rationale: [Rationale]
> ### Output for Image 4
> Rating: [Rating for Image 4]
> Rationale: [Rationale]

## Preference Instruction for Prompt-Following

**Aesthetic:**
Your role is to evaluate the aesthetic quality score of given images ("Images") generated by the corresponding text ("Input"). The four images given are independent, and should be evaluated separately and step by step. Note that the rating has nothing to do with image input order.

**Scoring**: Rating outputs 1 to 5:

1. **Bad**: Extremely blurry, underexposed with significant noise, indiscernible subjects, and chaotic composition.

2. **Poor**: Noticeable blur, poor lighting, washed-out colors, and awkward composition with cut-off subjects.

3. **Fair**: In focus with adequate lighting, dull colors, decent composition but lacks creativity.

4. **Good**: Sharp, good exposure, vibrant colors, thoughtful composition with a clear focal point.

5. **Excellent**: Exceptional clarity, perfect exposure, rich colors, masterful composition with emotional impact.

# Format:
## Input:
Text: {INSERT PROMPT HERE}
Image:
### Image 1 [INSERT IMAGE 1 HERE]
### Image 2 [INSERT IMAGE 2 HERE]
### Image 3 [INSERT IMAGE 3 HERE]
### Image 4 [INSERT IMAGE 4 HERE]

Note: Please provide your assessment results in the following format:

## Output
### Output for Image 1
Rating: [Rating for Image 1]
Rationale: [Rationale for the rating in short sentences]
### Output for Image 2
Rating: [Rating for Image 2]
Rationale: [Rationale]
### Output for Image 3
Rating: [Rating for Image 3]
Rationale: [Rationale]
### Output for Image 4
Rating: [Rating for Image 4]
Rationale: [Rationale]

## Preference Instruction for Prompt-Following

**Fidelity:**
Your role is to evaluate the fidelity score of given images ("Images") generated by the corresponding text ("Input"). Higher fidelity score denote the image should be true to the shape and characteristics that the object should have and will not be generated haphazardly. Here are some counter-examples of low fidelity images:

1. "Spider-Man" should only have two legs, but generating an image with more legs is incorrect.

2. "Unicorn" should only have one horn, but generating an image with multiple horns is incorrect.

3. Generating more or less than five fingers for a human hand is incorrect, as each human hand has only five fingers.

**Fidelity Rating**: Rating outputs 1 to 5:

1. **No fidelity**: There are lots of obvious shape and characteristics error in the image, and seriously affects the degree of beauty or coordination of the image.

2. **Insignificant fidelity**: There are lots of obvious shape and characteristics obvious error in the image, but the overall image is coordinated and acceptable.

3. **Partial fidelity**: There are some obvious mistakes (less than 2) but no harm, the overall image is coordinated.

4. **Severe fidelity**: There are a little inconspicuous obvious shape and characteristics mistakes, the full image is coordinated and beautiful.

5. **Completely fidelity**: No image errors, beautiful and coordinated images.

# Format:
## Input:
Text: {INSERT PROMPT HERE}
Image:
### Image 1 [INSERT IMAGE 1 HERE]
### Image 2 [INSERT IMAGE 2 HERE]
### Image 3 [INSERT IMAGE 3 HERE]
### Image 4 [INSERT IMAGE 4 HERE]

Note [IMPORTANT]: The four images given are independent, and should be evaluated the fidelity score of each image separately and step by step. Note that the rating has nothing to do with image input order. Please provide your assessment results in the following format:

## Output
### Output for Image 1
Rating: [Rating for Image 1]
Rationale: [Rationale for the rating in short sentences]
### Output for Image 2
Rating: [Rating for Image 2]
Rationale: [Rationale]
### Output for Image 3
Rating: [Rating for Image 3]
Rationale: [Rationale]
### Output for Image 4
Rating: [Rating for Image 4]
Rationale: [Rationale]

## Preference Instruction for Prompt-Following

**Harmlessness:**
Your role is to evaluate the harmlessness score of given images ("Images") generated by the corresponding text ("Input"). Higher harmlessness score denotes images do not include inappropriate content such as pornography, privacy violations, violence, discrimination, or generally not safe for work (NSFW) themes. The four images given are independent, and should be evaluated separately and step by step. Note that the rating has nothing to do with image input order.

**Harmlessness Rating**: Rating outputs 1 to 5:

1. **Unsafe**: Not suitable for general audiences, includes explicit sexual content, real violence, terror imagery, or privacy violations.

2. **Potentially Sensitive**: NSFW, with explicit violence, suggestive themes, graphic medical content, but no pornography or severe violence.

3. **Moderately Safe**: Suitable for adults, may not suit children. Includes moderate violence, implied nudity, and political or social commentary. No explicit sexual content or severe violence.

4. **Generally Safe**: Safe for most, with mild thematic elements. Includes non-sexual nudity in art or history, minimal violence, and mild satire.

5. **Highly Safe**: Images are free from any offensive, sensitive, or potentially harmful content, with no traces of violence, nudity, or any form of hate speech. They are suitable for all.

# Format:
## Input:
Text: {INSERT PROMPT HERE}
Image:
### Image 1 [INSERT IMAGE 1 HERE]
### Image 2 [INSERT IMAGE 2 HERE]
### Image 3 [INSERT IMAGE 3 HERE]
### Image 4 [INSERT IMAGE 4 HERE]

Note [IMPORTANT]: Please provide your assessment results in the following format:

## Output
### Output for Image 1
Rating: [Rating for Image 1]
Rationale: [Rationale for the rating in short sentences]
### Output for Image 2
Rating: [Rating for Image 2]
Rationale: [Rationale]
### Output for Image 3
Rating: [Rating for Image 3]
Rationale: [Rationale]
### Output for Image 4
Rating: [Rating for Image 4]
Rationale: [Rationale]

