# OpenReview forum: "Multimodal Large Language Models Make Text-to-Image Generative Models Align Better"
_NeurIPS.cc/2024/Conference — NeurIPS 2024 poster_

### Official Review · Reviewer_8ytx · 2024-06-15

**Soundness:** 3
**Presentation:** 4
**Contribution:** 4
**Rating:** 6
**Confidence:** 3

**Summary:**

This paper introduces VisionPrefer, a large-scale, high-quality, and fine-grained preference dataset for text-to-image generative alignment.  VisionPrefer offers advantages in scalability, fine-grained annotations, and a comprehensive feedback format compared with existing preference datasets. The authors further propose a reward model, VP-Score, which exhibits competitive correlation with human preferences by utilizing VisionPrefer. The experimental results underscore the effectiveness of both VisionPrefer and VP-Score.

**Strengths:**

1.  The idea that utilizing a MLLM as a human-aligned preference annotator for text-to-image generation sounds reasonable.

2. The introduced dataset VisionPrefer contains multiple preference aspects including prompt-following, aesthetic, fidelity, and harmlessness, which can contribute to generating images that are more aligned with human preferences.

3. The fine-tuned text-to-image generation model achieves enhanced performance in various aspects from fine-grained feedback.

4. The content of this paper is substantial, offering a wealth of experiments and analyses.

**Weaknesses:**

1. This paper demonstrates that fine-grained feedback from MLLM help to yield more human-preferred images in terms of prompt-following, esthetic, fidelity and harmlessness with qualitative results.  It is better if the authors could provide some quantitative metrics.

2. A significant contribution of this paper lies in the dataset. However, examples of annotations in VisionPrefer can only be found in the appendix. It would be more intuitive if these were included in the main paper.

**Questions:**

Please see the wearknesses.

**Limitations:**

The authors have discusses the limitations in the paper.

---

> ### Author Rebuttal · Authors · 2024-08-05
>
> Thank you for your detailed, helpful feedback. We address your feedback point by point below.
>
> ---
>
> > **Q1**: It is better if the authors could provide some quantitative metrics.
>
> **A1**: Thank you very much for your insightful feedback :). Firstly, in **Table 2 of the main text**, we provide a comparison of the preference prediction accuracy scores of our reward model VP-Score, with other state-of-the-art reward models. Additionally, we present the numerical results for win count & win rate under PPO and DPO experiments in **Tables 4 and 5 in the appendix**. These quantitative results demonstrate the effectiveness of using MLLMs to generate preference feedback.
>
> Moreover, in response to your valuable suggestion, we conducted additional measurements. We compared the FID values (a widely recognized metric for evaluating the quality of images generated by models) under PPO and DPO experimental settings for generative model optimized with our data and model against those optimized with other models. As shown in **Tables A and B**, we found that the FID scores of ours are significantly lower than those of the other comparison groups, indicating the effectiveness of using MLLMs-generated feedback to align the generation model.
>
> We will include these findings in the revised version of our paper to enrich the content. If you have other quantitative metrics of interest, please feel free to let us know :).
>
> ***Table A.** FID score of generative models optimized with VP-Score compared to other reward models for PPO experiments.*
> |                     | FID Score $\downarrow$ |
> | ------------------- | ---------------------- |
> | CLIP                | 8.32                   |
> | Aesthetic           | 8.17                   |
> | ImageReward         | 8.44                   |
> | PickScore           | 7.62                   |
> | HPS v2              | 7.51                   |
> | **VP-Score (Ours)** | **7.44**               |
>
> ***Table B.** FID score of generative models optimized with VisionPrefer compared to other human preference datasets for DPO experiments.*
> |                         | FID Score $\downarrow$ |
> | ----------------------- | ---------------------- |
> | ImageRewardDB           | 8.97                   |
> | HPD                     | 7.70                   |
> | Pick-a-Pic              | 6.62                   |
> | **VisionPrefer (Ours)** | **6.43**               |
>
> ---
>
> > **Q2**: A significant contribution of this paper lies in the dataset. However, examples of annotations in VisionPrefer can only be found in the appendix. It would be more intuitive if these were included in the main paper.
>
> **A2**: Considering your insightful recommendations, in the revised paper, we will move the examples of annotations in VisionPrefer to the main body to make the central focus of the article clearer and more reader-friendly :).

---

> > ### Comment · Reviewer_8ytx · 2024-08-10
> >
> > Thans for the authors' response. I tend to keep my original rating for acceptance.

---

> > > ### Comment · Area_Chair_ayqa · 2024-08-13
> > >
> > > Dear reviewer,
> > >
> > > thanks for your participation in the NeurIPS peer review process.  Thanks for indicating that you are leaning towards accepting the paper.  Is the response from authors satisfactory? Does it address weaknesses that you mentioned in the review?
> > > - If yes, are you planning to increase your score?
> > > - If no, could you help the authors understand how they can improve their paper in future versions?
> > >
> > > Thanks,
> > > AC

---

### Official Review · Reviewer_AdFm · 2024-07-03

**Soundness:** 3
**Presentation:** 3
**Contribution:** 3
**Rating:** 7
**Confidence:** 4

**Summary:**

The paper introduces VisionPrefer, a large-scale, fine-grained preference dataset constructed using multimodal large language models (MLLMs) as annotators. VisionPrefer aims to improve the alignment of text-to-image generative models with human preferences by providing detailed feedback on generated images. The dataset is created through a three-step process: generating and polishing prompts, generating images using various models, and obtaining preference annotations from GPT-4 V, including scalar scores, rankings, and textual explanations.

**Strengths:**

VisionPrefer introduces an innovative approach to generating a fine-grained preference dataset by leveraging multimodal large language models (MLLMs) as annotators. The combination of prompt generation, diverse image generation, and detailed preference annotations is novel. The use of GPT-4 V to provide multi-faceted feedback, including scalar scores, rankings, and textual explanations, distinguishes VisionPrefer from existing preference datasets that rely solely on human annotations. This approach not only automates the annotation process but also introduces a level of detail and consistency that is challenging to achieve with human annotators alone.

The paper is well-organized and clearly written. The structure follows a logical flow, from the introduction and motivation to the methodology, experiments, and analysis.

VisionPrefer has significant implications for the field of text-to-image generation. The idea that using MLLM to guide the alignment ofimage generation models is interesting.

**Weaknesses:**

Convergence Speed of VP-Score Figure 3 indicates that VP-Score converges significantly slower than other baselines. This slower convergence raises questions about the efficiency and practicality of using VP-Score in real-world applications. The authors should investigate and explain the reasons behind this slower convergence. Potential factors could include the complexity of the fine-grained annotations or the training dynamics of the reward model. Providing insights into this issue and suggesting potential optimizations to improve convergence speed would enhance the utility of VP-Score.


Inconsistent Improvement Across Datasets
Despite the larger scale and richer annotations of VisionPrefer, the dataset does not achieve consistent improvements across all evaluated datasets. This inconsistency suggests that VisionPrefer may not fully capture human preferences, indicating potential limitations in the design of the annotations or the preference modeling approach. The authors should analyze why VisionPrefer fails to deliver uniform performance gains and identify aspects where it falls short. This analysis could involve examining the nature of the datasets where VisionPrefer underperforms and exploring ways to refine the annotation process to better align with human preferences. Additionally, providing more detailed comparisons and breakdowns of performance across different datasets would offer valuable insights into the strengths and weaknesses of VisionPrefer.

**Questions:**

see weaknesses

**Limitations:**

Bias in Annotations: The use of AI-generated annotations (e.g., GPT-4 V) could propagate existing biases, leading to biased outputs from generative models. The authors should discuss mechanisms to detect and mitigate such biases.

Combining AI and Human Annotations: Explore the benefits of combining AI and human annotations to enhance the diversity and generalizability of preference data, ensuring the model aligns more closely with a broad range of human preferences.

---

> ### Author Rebuttal · Authors · 2024-08-05
>
> Thank you for your detailed, helpful feedback. We address your feedback point by point below.
>
> ---
> >**Q1**: Figure 3 indicates that VP-Score converges slower than others.
>
> **A1**: In fact, **Figure 3 does not represent a comparison of the convergence speeds of VP-Score and other baselines**. Instead, it illustrates how the human preference quality of a generative model, which uses VP-Score as a reward model for alignment, changes during the fine-tuning process (as noted in lines 173-177 of the main text). This is assessed by various reward models such as ImageReward and HPS v2.
>
> As shown in Figure 3, the quality of human preferences in the generative model increases as the fine-tuning steps progress. This improvement is reflected in the increasing reward score of models such as HPS v2, which are trained on human preference data. **This indicates that VP-Score can serve as a reliable reward model for aligning generative models, further proving the reliability of using MLLMs to generate human preference feedback.**
>
> Additionally, inspired by your feedback, we observed that the growth in metrics for ImageReward, PickScore, and Aesthetic is not very pronounced. We hypothesize that these three models may struggle to capture variations in the quality of human preferences in images. In contrast, HPS v2, currently the most accurate and unbiased model for reflecting human preference quality, exhibits a clear upward trend. **This suggests that our VP-Score aligns better with HPS v2, further validating the effectiveness of VP-Score.**
>
> Furthermore, we appreciate your suggestion. All authors agree that the convergence speed of using human preference reward models for aligning generative models is an intriguing research question, and we plan to explore this in future work.
>
> ---
> >**Q2**: The authors should analyze why VisionPrefer fails to deliver uniform performance gains and identify aspects where it falls short.
>
> **A2**: We first analyzed the differences between GPT-4V and human annotations at a fine-grained level: we randomly selected 1,000 samples from VisionPrefer and invited six human experts to rate them on four dimensions (Prompt-Following, Aesthetic, Fidelity, and Harmlessness). The correlation between human experts' preference judgments and GPT-4V's judgments is shown in Table A below. We found that GPT-4V's preference annotations align more closely with human experts in the areas of Prompt-Following and Fidelity (the same phenomenon can be observed in Figure 10 e in the main text). **This suggests that GPT-4V is better aligned with human preferences in these two aspects**. This finding might indicate that we can reduce the weight of Aesthetic and Harmlessness scores when constructing VisionPrefer to achieve more accurate preference annotations.
>
> Furthermore, **we hypothesize that using the average score across the four aspects as the final preference score in VisionPrefer might differ from human preference evaluations**. Humans may prioritize certain aspects more heavily, such as Prompt-Following and Aesthetic, which could be more significant than Harmlessness. Supporting this hypothesis, Table 2 in the main text shows that VP-Score achieves the highest accuracy on ImageRewardDB but not on the other two datasets. ImageRewardDB uses an averaged score across multiple aspects for human preference evaluation, similar to our VisionPrefer construction, while Pick-a-Pic and HPD involve direct human preference scoring without fine-grained evaluation across multiple aspects.
>
> Based on these insights, we envision two possible improvements for GPT-4V annotations: (1) Require GPT-4V to provide a confidence estimate for each aspect-specific preference score, using a weighted average of the four aspects as the final sample score. (2) Modify GPT-4V’s annotation process to evaluate all four aspects simultaneously for each text-image sample, outputting a final score rather than an average.
>
> In summary, your question is very intriguing, and we will continue to explore and experiment with these ideas in future research.
>
> ***Table A.***
>
> |Prompt-Following|Aesthetic|Fidelity|Harmlessness|
> |-|-|-|-|
> |94.1%|73.7%|92.0%|79.3%|
>
> ---
> >**Q3**: Bias in Annotations.
>
> **A3**: All authors consider your suggestion both important and intriguing. In response, we have reviewed and searched for related research [1-4], including analyses of biases in AI-generated annotations and methods for detecting them. We will focus on these issues in future work to further optimize our VisionPrefer and VP-Score. Additionally, we will include your suggestion in the **Limitations** section to enrich the paper.
>
> **Reference**
>
> [1] Fang, Xiao, et al. "Bias of AI-generated content: an examination of news produced by large language models.".
>
> [2] Fan, Zhiting, et al. "BiasAlert: A Plug-and-play Tool for Social Bias Detection in LLMs.".
>
> [3] Morehouse, Kirsten, et al. "Bias Transmission in Large Language Models: Evidence from Gender-Occupation Bias in GPT-4.".
>
> [4] Hajikhani, Arash, and Carolyn Cole. "A critical review of large language models: Sensitivity, bias, and the path toward specialized ai.".
>
> ---
> >**Q4**: Combining AI and Human Annotations.
>
> **A4**: Thanks. In response, we conducted the following experiment: we combined our VisionPrefer dataset (AI annotations) with a randomly selected subset from the Pick-a-Pic dataset (human annotations). We then retrained a new reward model, named VP-Score+x, where x represents the percentage of Pick-a-Pic data used, while keeping the training steps constant. The preference prediction accuracy scores are shown in Table B below.  **We found that mixing AI and human annotations further improves the reward model's accuracy in reflecting human preferences**.
>
> ***Table B.***
>
> |Model|ImageRewardDB|HPDv2|Pick-a-Pic|
> |-|-|-|-|
> |PickScore|62.9|79.8|70.5|
> |HPSv2|65.7|83.3|67.4|
> |VP-Score|66.3|79.4|67.1|
> |VP-Score+10|66.7|79.7|69.0|
> |VP-Score+20|66.9|80.2|69.5|
> |VP-Score+30|**67.1**|**80.3**|**70.7**|

---

### Official Review · Reviewer_zspv · 2024-07-13

**Soundness:** 2
**Presentation:** 3
**Contribution:** 3
**Rating:** 4
**Confidence:** 3

**Summary:**

The paper presents a new AI-generated dataset aimed at enhancing text-to-image generative models by aligning them more closely with human preferences. The data is annotated by multimodal large language models (MLLMs) and captures detailed preferences across multiple dimensions like prompt-following, aesthetic, fidelity, and harmlessness. The dataset facilitates the training of a reward model called VP-Score, which approximates human preference predictions.

**Strengths:**

1. it introduces a detailed dataset for text-to-image generation preferences.

2. the research has involved human study to validate the results, showing the strength of the results.

3. the experiments section has multiple elements, including results allowing editing of the generated images.

**Weaknesses:**

1. the overall novelty is a bit limited as it heavily relies on existing large models to generate the dataset and to train additional models (more in the questions).

2. the paper seems to have a diverse set of focuses. For example, the section regarding editing images does not seem to correlate well with the main proposed method, thus seem irrelevant.

**Questions:**

1. The biggest question would be the relevance of the proposed method. The authors only showed the empirical strength of the proposed method with a randomly sampled subset of the images. In this case, the true value of the proposed method can hardly be validated. I would highly recommend the authors to repeat the experiments with the standard large-scale training regime. I understand the computation requirement will be huge, but there does not seem to have an alternative path to show the power of the proposed method.

2. Similarly as above, a big question would be whether the proposed method is only due to the power of GPT-4V, which is a larger model, to show the true value of the method, the authors might have to use the method and GPT-4V to train a bigger model.

**Limitations:**

The paper has an analysis section and ablation study in the appendix. An explicit limitation section can further help readers appreciate the paper.

---

> ### Author Rebuttal · Authors · 2024-08-05
>
> Thank you for your feedback. We address your questions point by point below:
>
> ---
> >**Q1**: the overall novelty is a bit limited as it heavily relies on existing large models to generate the dataset and to train additional models.
>
> **A1**: In fact, the main motivation for our research is to explore: **Can Multimodal Large Language Models act as a Human-Aligned Preference Annotator for Text-to-Image Generation?**  (as noted in line 38 of the main text). **We aim to demonstrate that MLLMs can generate reliable human preference annotations for text-to-image generation results**. To this end, we explored the performance of several state-of-the-art MLLMs (GPT-4V, Gemini-pro-V, and LLaVA 1.6) in generating human preference annotations (see Figure 10a in the main text). Using the best-performing GPT-4V, we constructed VisionPrefer, the largest fine-grained preference dataset to date, and developed the reward model VP-Score derived from VisionPrefer. We validated the effectiveness of VisionPrefer and VP-Score under PPO and DPO experimental settings.
>
> We believe our research has the following significance:
>
> * Since human annotation of preference labels is expensive and time-consuming, proving the reliability of MLLMs in generating human preference annotations allows humans to easily scale existing human preference datasets, thereby achieving better text-to-image generative alignment.
> * It further supports the notion that AI-generated synthetic data is a reliable way to create augmented datasets and provides a large dataset for further research related to AI-generated synthetic data.
> * It shows that existing AI models may exhibit capabilities similar to those of human experts in some interesting areas, e.g., image aesthetics, which is exciting and worthy of further exploration.
>
> ---
> >**Q2**: the paper seems to have a diverse set of focuses. For example, the section regarding editing images does not seem to correlate well with the main proposed method, thus seem irrelevant.
>
>
> **A2**: We included the section on editing images to demonstrate that using MLLMs for preference annotation offers more than just aligning generative models to produce images that match human preferences, similar to existing human-annotated preference datasets. **A key advantage of MLLMs is their ability to provide detailed textual feedback for each text-to-image sample, which can guide further image editing to produce images that better align with human preferences.** This is another critical advantage of MLLMs beyond their ability to construct reliable preference labels quickly and cost-effectively. In this section, to further validate this advantage, we designed a simple pipeline (see in the Figure 8 in the maintext) using MLLM-generated textual feedback for image editing. Therefore, this section does not indicate a lack of focus in our paper but rather highlights the advantages of our approach.
>
> ---
> >**Q3**: The authors only showed the empirical strength of the proposed method with a randomly sampled subset of the images. I recommend the authors to repeat the experiments with the standard large-scale training regime.
>
> **A3**: We demonstrated the effectiveness of our conclusions and methods in two ways:
>
> * For the accuracy of the reward model's preference prediction (see Table 2 in the main text), we computed quantitative metrics on three public test sets without any random components, ensuring reliable results.
> * To validate the model alignment under PPO and DPO settings, we followed existing research [1]. We generated 64 images for all aligned generative models and selected the top 3 images using the corresponding reward model. We then invited **ten human annotators** to evaluate these top 3 images. **This process was repeated on three test sets for both PPO and DPO settings**, with a workload and rigor that exceeds previous studies [1] (only three annotators on one test set). Therefore, we believe the qualitative and quantitative results presented in this paper substantiate the validity of our conclusions and methods.
>
> Unfortunately, we did not fully understand your suggestion to "repeat the experiments with the standard large-scale training regime." We look forward to further discussions during the diffusion stage and welcome more specific feedback to conduct additional experiments.
>
> **Reference**
>
> [1] Xu, Jiazheng, et al. "Imagereward: Learning and evaluating human preferences for text-to-image generation." NeurIPS 2024.
>
> ---
> >**Q4**: To show the true value of the method, the authors might have to use the method and GPT-4V to train a bigger model.
>
> **A4**: The primary motivation of our research is to explore: **Can Multimodal Large Language Models act as a Human-Aligned Preference Annotator for Text-to-Image Generation?** We aim to demonstrate that MLLMs can generate reliable human preference annotations for text-to-image generation results. Proving that MLLMs can reliably generate these annotations offers many benefits, such as constructing preference labels quickly and cost-effectively.
>
> Therefore, we naturally rely on the powerful capabilities of MLLMs (such as GPT-4V), as stronger models may produce higher-quality preference annotations, which is a desirable outcome. Unfortunately, we did not fully understand your specific suggestion regarding "training a bigger model to show the true value of the method." We look forward to further discussions with you during the diffusion stage and are eager to conduct additional experiments based on these more detailed discussions.
>
> ---
> >**Q5**: Require for an explicit limitation section.
>
> **A5**: Thank you for your suggestion. We will include an explicit limitation section in the latest version, summarized as follows: Due to resource constraints, although our VisionPrefer is the largest fine-grained preference dataset for text-to-image generative alignment, we believe it can be further scaled to achieve better alignment performance. We plan to expand it further in the future.

---

> ### Comment · Area_Chair_ayqa · 2024-08-13
> **Reviewer zspv: please respond to the authors' rebuttal**
>
> Dear reviewer,
>
> thanks for your participation in the NeurIPS peer review process.  We are waiting for your response to the rebuttal.  You gave a reject rating (3), with 2 weaknesses and 2 questions.
>
> Is the response from authors satisfactory?
> - If yes, are you planning to increase your score?
> - If no, could you help the authors understand how they can improve their paper in future versions?
>
> Thanks,
> AC

---

> > ### Comment · Reviewer_zspv · 2024-08-13
> > **Replies to rebuttal**
> >
> > Dear authors,
> >
> > Thank you for offering a comprehensive rebuttal in a polite manner despite my review is the only one showing negative polarity. I understand you have concerns about the review; sharing the same concerns, I have re-read the paper, which is why it takes time for me to respond.
> >
> > First of all, by the question raised "Unfortunately, we did not fully understand your specific suggestion regarding 'training a bigger model to show the true value of the method.'" I was referring to the experiment in section 4.2, which I believe is to use the constructed dataset to show that generative models can benefit from the dataset to align better (per line 157). In this case, it seems to me that the authors are forming a loop of construct a dataset with a bigger generative model and then show that it can help train a better smaller model. This is where I ask the question about the "true value" of the method yet to be validated.
> >
> > I understand the above requirement might be difficult to implement but I believe it will be essential for the study. For example, a simple logic loophole is that the bigger model will of course be a better aligned one and help the smaller model to perform better, it's like the conventional knowledge distillation research that we can get better small model with the help of a big model, this does not mean the same techniques will help improve the bigger model.
> >
> > On the other hand, the authors might want to respond that "getting a better big model" is not the goal of this research, this research is about investigating how GPT-4V level models can offer better-aligned data to train smaller model in the beginning. In this case, I agree the paper is better justified, but in this case, it seems to me the paper will need some re-writing work, e.g., line 19 sends out a much stronger message than this.
> >
> > Again, I thank the reviewers for offering a professional and polite rebuttal, and it takes me a while to respond because the current situation makes me feel like I need to re-read the paper, but it seems some of the concerns are still there. There is still a chance that I misread some parts, and I'm here for the authors to enlighten on these parts.

---

> > > ### Author Response · Authors · 2024-08-13
> > > **Response to Reviewer zspv**
> > >
> > > Dear Reviewer zspv:
> > >
> > > Thank you for your detailed feedback. We believe there are some misunderstandings that we need to clarify:
> > >
> > > 1. First, we want to clarify that the two types of models involved in this paper (MLLMs, specifically GPT-4V, and text-to-image generative models) are not the same type of models. The text-to-image generative model (simplified as T2I) takes text as input and produces images, while the language-based MLLMs take both text and images as input and output text. **Therefore, they are not the same kind of models as large and small models, and this differs from distillation**, which simply transfers the capabilities of a large model to a smaller model for the same task. **To consider this as distillation, MLLMs would need to provide reliable T2I preference labels. But is it certain that they are more reliable and stronger preference providers?** Although MLLMs are powerful tools proven to offer various functions (such as image captioning, QA, etc.), **there has been no prior research exploring or proving that MLLMs can produce reliable preference labels. Our research is the first to address this concern and demonstrate this point (which is the core motivation of our study, see line 39)**. Therefore, we used GPT-4V to create VisionPrefer and conducted a series of extensive experiments to show that MLLMs are indeed reliable preference providers. This is one of the main contributions and value propositions of our paper.
> > > 2. **We want to emphasize that our approach and the research motivation of this paper are independent of and unrelated to the size of the T2I models.** Preference labels reflect human preferences for different images generated from the same prompt. We have demonstrated that, for current-scale T2I models, the preference labels provided by MLLMs are effective (sections 4) and align with those provided by humans. Current-scale T2I models can further optimize their generation results by learning from these preference labels to better align with human preferences. **If very large-scale T2I models (perhaps on the same scale as GPT-4V) learn little from these preference labels, this would occur with both GPT-4V-generated and human-annotated preference labels.** This does not affect the value of our research; rather, **it raises the question of whether aligning very large-scale T2I models is necessary, which is an interesting research topic in its own right** (though currently difficult to verify).
> > > 3. **We selected the SD v1.5 model following previous research settings in the T2I alignment domain [1-5].** Your suggestion to align larger T2I models to verify the effect is an interesting proposal that could enrich our research. However, we believe that the existing timeline is insufficient to support further experimental feedback. We plan to conduct experiments in this area in the future.
> > > 4. Lastly, we want to reiterate that the research motivation of this paper is to explore "Can Multimodal Large Language Models act as a Human-Aligned Preference Annotator for Text-to-Image Generation?" (line 39). We have emphasized the research motivation of our paper in many places throughout the text (e.g., lines 20, 39, 200, 240, 288, etc.). To avoid any remaining misunderstandings, we will make sure to revise the paper to highlight our motivation more clearly in the modified version.
> > >
> > > ---
> > >
> > > Additionally, we would like to reiterate the broader value of our work, which extends beyond just VisionPrefer and VP-Score:
> > >
> > > 1. Since human annotation of preference labels is expensive and time-consuming, proving the reliability of MLLMs in generating human preference annotations allows humans to easily scale existing human preference datasets, thereby achieving better text-to-image generative alignment.
> > > 2. It further supports the notion that AI-generated synthetic data is a reliable way to create augmented datasets and provides a large dataset for further research related to AI-generated synthetic data.
> > > 3. It shows that existing AI models may exhibit capabilities similar to those of human experts in some interesting areas, e.g., image aesthetics, which is exciting and worthy of further exploration.
> > >
> > > ---
> > >
> > > **If you have any further questions or concerns, please feel free to contact us at any time. We are always available and look forward to further discussions with you. :)**
> > >
> > > Best regards,
> > >
> > > All Authors
> > >
> > > **Reference**
> > >
> > > [1] Xu, Jiazheng, et al. "Imagereward: Learning and evaluating human preferences for text-to-image generation." NeurIPS 2024.
> > >
> > > [2] Fan, Ying, et al. "Reinforcement learning for fine-tuning text-to-image diffusion models." NeurIPS 2024.
> > >
> > > [3] Wallace, Bram, et al. "Diffusion model alignment using direct preference optimization." CVPR 2024.
> > >
> > > [4] Yang, Kai, et al. "Using human feedback to fine-tune diffusion models without any reward model." CVPR 2024.
> > >
> > > [5] Clark, Kevin, et al. "Directly fine-tuning diffusion models on differentiable rewards." ICLR 2024.

---

### Decision · Program_Chairs · 2024-09-25

**Decision:**

Accept (poster)

**Comment:**

Summary of Review Process:
- 3 reviews.
- Mixed scores: 4,6,7
- Authors submitted rebuttals/responses
- 1 out of 3 reviewers did not engage in discussion


Meta Review:
There was consensus from all reviewers regarding the novelty of the approach, clarity of writing, and experimental findings of the paper.  The following is a summary of questions and rebuttal:
- Reviewer AdFm who did not participate in the discussion gave an "Accept" (7) rating with two weaknesses mentioned: (1) efficiency/practicality concerns due to slow convergence (2) inconsistent improvement across datasets. The reviewer also pointed out two limitations.
    - The authors have added some clarification for (1) and have acknowledged that improving the convergence speed should be a future research direction.
    - The authors will be adding a note about bias in annotations to the Limitations sections.
    - The authors have provided more insight into the failures of the VisionPrefer model along with additional experiments for combining human and AI annotations.
- Reviewer 8ytx mentioned two weaknesses/requests: (1) provide quantitative metrics and (2) move details about the dataset to the main paper
    - authors have responded to (1) by stating that some quantitative metrics are in Table 2 of the main text and Tab 4,5 in Appendix. Authors also performed additional measurements and compared FID values, to be included in the final version
- Reviewer zspv mentioned 4 weaknesses/questions. Q1 was about novelty, Q2 was about the focus of the paper, Q3 was about experimental setup, and Q4 was about training a bigger model.
    - authors and reviewers discussed these concerns but there was no change in the rating as there was a disagreement on the  focus and scope of the paper

Overall, the AC believes this to be an interesting work on leveraging MLLMs to create a preference dataset and training a reward model to guide the tuning of text-to-image generative models. The AC's recommendation is to carefully integrate suggestions from reviewers and additional experiments/evidence/clarifications that emerged from the discussion, into the final version.